# Combining genomics and epidemiology to investigate a zoonotic outbreak of rabies in Romblon Province, Philippines

Mirava Yuson [1,2,8] ✉, Criselda T. Bautista[1,3,8], Eleanor M. Rees[1],
Carlijn Bogaardt[4], Van Denn D. Cruz[2], Rowan Durrant[1], Anna Formstone [1],
Daria L. Manalo[3], Duane R. Manzanilla[2], Mikolaj Kundergorski [1], Leilanie Nacion[3],
Hannaniah Aloyon[3], Jude Karlo Bolivar[3], Jeromir Bondoc[3], Christina Cobbold[5],
Efraim Panganiban[3], Shynie Vee. M. Telmo[6], Jobin Maestro[7],
Mary Elizabeth G. Miranda[2], Nai Rui Chng[1], Kirstyn Brunker [1,8] &
Katie Hampson [1,8]

Rabies is a viral zoonosis that kills thousands of people annually in low- and middle-income countries across Africa and Asia where domestic dogs are the reservoir. 'Zero by 30', the global strategy to end dog-mediated human rabies, promotes a One Health approach underpinned by mass dog vaccination, post-exposure vaccination of bite victims, robust surveillance and community engagement. Using Integrated Bite Case Management (IBCM) and whole genome sequencing (WGS), we enhanced rabies surveillance to detect an outbreak in a formerly rabies-free island province in the Philippines. We inferred that the outbreak was seeded by at least three independent human-mediated introductions that were identified as coming from neighbouring rabies-endemic provinces. Considerable local transmission went undetected, and two human deaths occurred within 6 months of outbreak detection. Suspension of routine dog vaccination due to COVID-19 restrictions likely facilitated rabies spread from these introductions. Emergency response, consisting of awareness measures, and ring vaccination, were performed, but swifter and more widespread implementation is needed to contain and eliminate the outbreak and to secure rabies freedom. We conclude that strengthened surveillance making use of new tools such as IBCM, WGS, and rapid diagnostic tests can support One Health in action and progress towards the 'Zero by 30' goal.

Neglected tropical diseases persist in low- and middle-income countries (LMICs), causing major economic losses, morbidity and mortality[1]. Treatment and elimination prospects are limited by the inequitable allocation of financial resources, resulting in high morbidities affecting over one billion people worldwide. Economical strategies consisting of case-finding based on observed signs and history-taking, have been used for the epidemiological investigation of outbreaks of neglected tropical diseases including dengue fever[2] and leprosy[3]. Genomic surveillance has also proven valuable for tackling zoonotic disease emergence, including its application to outbreaks of Ebola[4], Lassa fever[5], Influenza[6], and Mpox[7], providing insights into transmission dynamics and the impacts of

interventions, and therefore informing more targeted control and prevention.

Rabies is an example of a neglected zoonotic disease caused by the rabies virus (RABV). It has long been a significant public health issue, and although the disease has been eliminated from several regions over the last century, rabies still kills thousands of people annually in Africa and Asia[8], where free-roaming dogs are common[9]. Despite being preventable through dog vaccination, rabies is re-emerging across much of Southeast Asia, including in Malaysia[10], Indonesia[11], and Vietnam[12]. The historically rabies-free Timor-Leste reported its first human death due to rabies in early 2024, highlighting the ongoing challenge of spread across the region[13]. Effective control of zoonoses like rabies requires a One Health approach with coordination between human and animal health sectors[14].

Rabies is fatal once symptoms appear, but progression to disease can be prevented if immediate post-exposure prophylaxis (PEP) is given to bite victims after exposure. PEP, while highly effective, should be part of a broader rabies management strategy which includes educational campaigns to increase awareness, robust surveillance for case detection, and mass dog vaccination to interrupt dog-to-dog transmission, thereby reducing the risk of human exposures[15]. Dog vaccination campaigns in the 20th century[16] led to the elimination of dog-mediated rabies in North America, Western Europe, and parts of Asia, and dramatically reduced cases across Latin America[17]. Similar measures have been applied at the community level in some rabies-endemic countries, leading to local rabies-free zones like Pontianak City in Indonesia[18] and N'Djaména in Chad[19]. However, introductions and re-emergence of rabies through animal importations by humans[20] or from natural incursions across borders occur regularly worldwide[21–26]. Examples from the city of Arequipa in Peru[27], Sarawak in Malaysia[10], and Mpumalanga province, South Africa[28], demonstrate how neglecting surveillance and dog vaccination can lead to rapid escalation from introductions in areas close to rabies-endemic zones. Inappropriate responses, such as dog culling, can also exacerbate spread, as seen in Indonesian islands Flores[29] and Bali[11], where failure to contain the epidemic led to enzootic transmission.

The hallmark of effective rabies control is strong intersectoral collaboration to reduce human mortality risks and eliminate disease from reservoir populations, which is why a One Health approach is recommended[30]. Incorporating a One Health approach has been shown to address common gaps in rabies surveillance, such as poor case detection[31]. Tools like rapid diagnostic tests (RDTs)[32–34], regular coordination between health workers from human and animal sectors through communication technologies[35,36], IBCM[37–40], and genomic sequencing[21,24,41] are known to offset surveillance weaknesses, while strengthening health systems for outbreak preparedness. IBCM is a rabies surveillance strategy that directly links public health and veterinary workers to manage animal bite incidents and prevent rabies[40]. It enhances surveillance through better case detection, improves patient care through more informed administration of PEP and can support better management of limited resources[42]. Sequencing of rabies viruses has identified new virus reservoirs[43], sources of introductions[44], and nearby populations that pose risks for re-emergence[21,31,45] and more generally improved our understanding of virus dispersal dynamics[46–48].

In 2015, the WHO launched the 'Zero by 30' global strategy to eliminate dog-mediated human rabies deaths by 2030[15]. However, achieving successful rabies control requires overcoming challenges such as limited human resources and cross-sectoral financing. Government priorities typically favour investment in animal diseases that have economic impacts such as African Swine Fever (ASF), whilst political and economic instability with frequent changes in governance make zoonotic disease control programmes difficult to maintain[49]. As a result, another major challenge to 'Zero By 30' is sustaining rabies freedom, as outbreaks reestablish due to lack of healthcare resources and siloing among health departments, undermining outbreak response.

Here, we report our learning from taking a One Health approach to tracking a rabies outbreak as it unfolded in a formerly rabies-free province in the Philippines. The investigation began with the initial detection of a rabid dog in 2022 on the island of Tablas, in Romblon Province (Fig. 1). The island had previously suffered from an incursion in 2011, and the ensuing outbreak caused 11 human fatalities (incidence of 3.48 deaths/100,000 persons/year) but no human or animal cases had been reported since 2012[23]. Our cross-sectoral and multi-disciplinary investigation used IBCM to enhance rabies surveillance as advocated by 'Zero by 30'[15] and deployed RDTs for early diagnosis. We further undertook WGS of rabies viruses from the outbreak to determine its probable origins and uncover the resulting spread.

## Results

### Rabies cases

Romblon province was considered rabies-free, with no cases recorded since 2012, until 2020 when two suspicious human deaths occurred in Romblon Island (Fig. 2A). Prior to 2012, four human deaths were recorded in the province between 2003 and 2006 and an outbreak on Tablas Island in 2011 confirmed eight animal cases and 11 human deaths[23]. From 2017 to 2021, rabies sample submissions steadily declined from 39 specimens tested annually to zero tested during lockdown. In late 2022, the use of IBCM identified a cluster of bite cases leading to the detection of the first dog rabies cases on Tablas Island, Romblon Province, in over a decade.

The first detected rabies-positive case (November 21st, 2022) was a dog that was investigated three days after its involvement in a biting incident (November 18th) in Santa Maria municipality. This was the first sample from the province to have been tested for rabies since 2020, and the first local use of an RDT after being supplied for IBCM (training carried out in March 2020 just before COVID-19 restrictions were announced). Due to the absence of laboratory facilities in Romblon province and the fluorescent microscope being broken at the Regional Animal Disease Diagnostic Laboratory (RADDL 4B, Fig. 1B), the sample was transported overnight to the National Reference Laboratory at the Research Institute for Tropical Medicine (RITM) in Manila. Here it was confirmed the next day (November 22nd) through direct fluorescent antibody testing (DFAT) and the positive result was immediately communicated to the local government, prompting increased sample collection, and in-field testing. That week two more samples collected from biting dogs in Odiongan and Alcantara municipalities were sent to RADDL 4B where they tested positive by RDT. Another biting dog from San Agustin municipality was classified as probable rabies after being killed and consumed without sample collection (Fig. 2). Positive confirmation of the first case in Santa Maria municipality prompted the sending of two frozen dog heads collected from Alcantara municipality in September and October 2022 to RITM in early December 2022. Both tested positive via DFAT, thus marking the index case of the outbreak as September 30th, 2022 (Fig. 2C).

Between September 2022 and September 2023, a total of 43 animal rabies cases and two human deaths were confirmed in eight out of the nine municipalities in Tablas Island (Fig. 2C). Additionally, three biting dogs were classified as probable cases, based on clinical signs[30] and progressive fatal outcomes consistent with those reported in literature[50,51], but without diagnostic confirmation due to lack of sample collection. The One Health link between public health and veterinary workers operationalized through IBCM was critical to identifying many of the rabid dogs ($n = 25$). Conversations on the IBCM peer support chat also disseminated information about rabid dogs that were investigated directly because of their strange behaviour (i.e. not because of biting a person). The Disease Surveillance Officer (appointed to coordinate IBCM across the provinces) facilitated resource sharing between sectors by transporting supplies, case

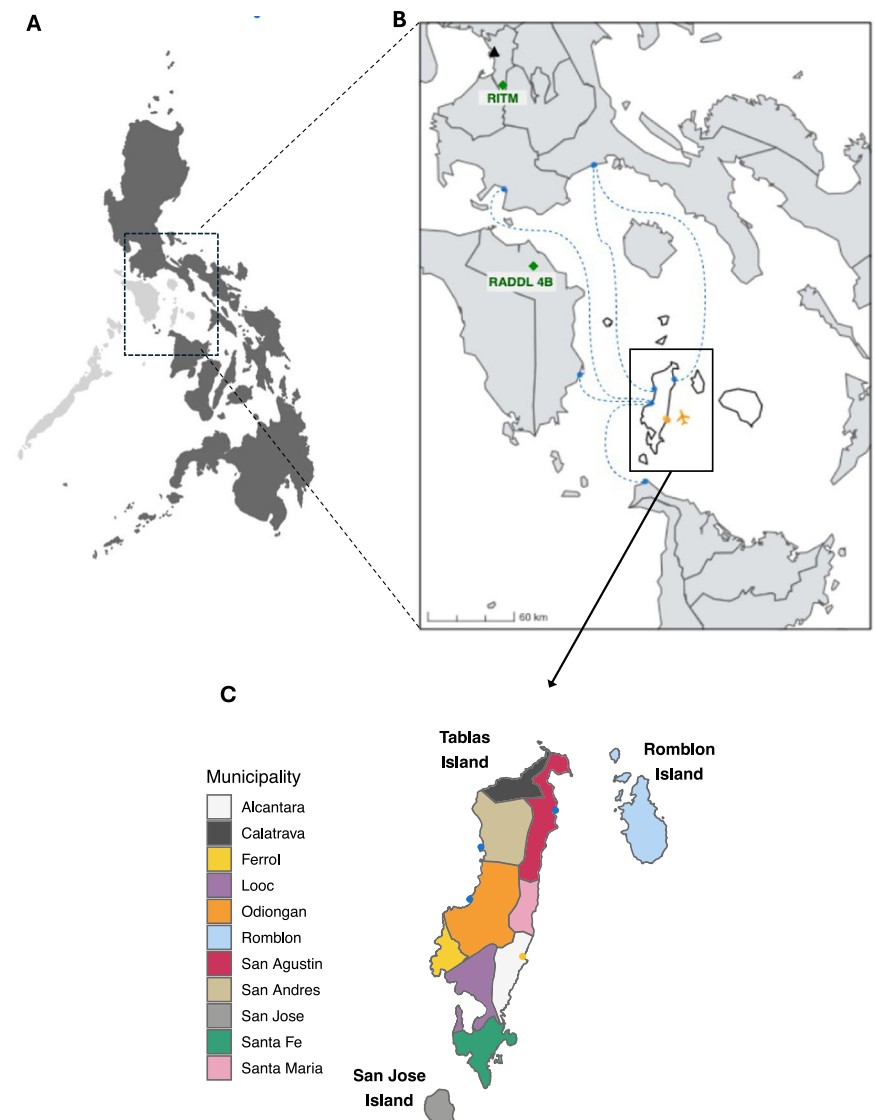

**Fig. 1 | Outbreak location in the formerly rabies-free province of Romblon in the Philippines. A** Location of MIMAROPA region, also known as Region 4B (light grey) within the Philippines, with the inset **B**) of Romblon Province (white) showing the Regional Animal Disease Diagnostic Laboratory (RADDL) and the Research Institute of Tropical Medicine (RITM) as green diamonds. Manila is indicated as a black triangle. Major ports are indicated in blue, the airport in yellow, and dashed blue lines show the main ferry routes to/from Tablas. **C** Tablas Island coloured by municipality, with the ports and airport coloured as above.

reports and vehicles so that investigations were conducted within 1–2 days of an animal death, before samples decomposed or became unfit for testing.

Of the submitted samples, 71.7% (43/60) tested positive for DFAT (all dogs). Only 3.3% (2/60) of submitted samples were from cats and both were negative. RDTs were used for initial screening of 51.7% (32/60) of samples. All RDT-positive samples were confirmed by DFAT. The RDT specificity was 100%, while sensitivity was 95.5%, with one initially negative RDT sample later confirmed positive by DFAT. Most positive cases were detected in San Agustin municipality (*n* = 13/43, 30.2%) where one of Tablas' ports is located (Fig. 1C). San Agustin had the highest sample submission rate, followed by Odiongan municipality, which accounted for 25.6% (11/43) of positive cases. No samples were collected from Ferrol municipality, nor were any probable rabid animals reported there. Most rabies-positive dogs were owned, while no owner could be identified for 32.6% of rabies-positive dogs (14/43). Twenty three point five percent (4/17) of rabies-negative dogs had a history of vaccination, while 13.6% (6/43)

of rabies-positive dogs had reportedly been vaccinated, although the vaccination year and the type of vaccine were unspecified, except for one of the dogs that became ill and died shortly after vaccination in 2023. Rabies-positive animals were either killed (27.9%), found dead (23.3%), died while under observation (41.9%), or had unspecified outcomes (7%).

Two human rabies deaths were identified in 2023: one in February (39 days after the bite in December 2022) and another in May (131 days after the bite) (Fig. 2B). The victims, a child from Santa Maria municipality and an older person from Odiongan municipality, were bitten by dogs and did not receive PEP. Initially, they sought treatment from local faith healers ('*tandok*'), as encouraged by their families and were hospitalised only when symptoms worsened.

On average, the delay between exposure and PEP in treated patients was 1.8 days (95% CI: 0.14–3.36 days; median of 0 days; *n* = 12). The mean delay between a biting incident and dog death in confirmed cases was 2.1 days (95% Confidence Interval (CI): 0.7–3.5 days), with a median delay of 1 day.

**A    2001 – 2023 Rabies cases**

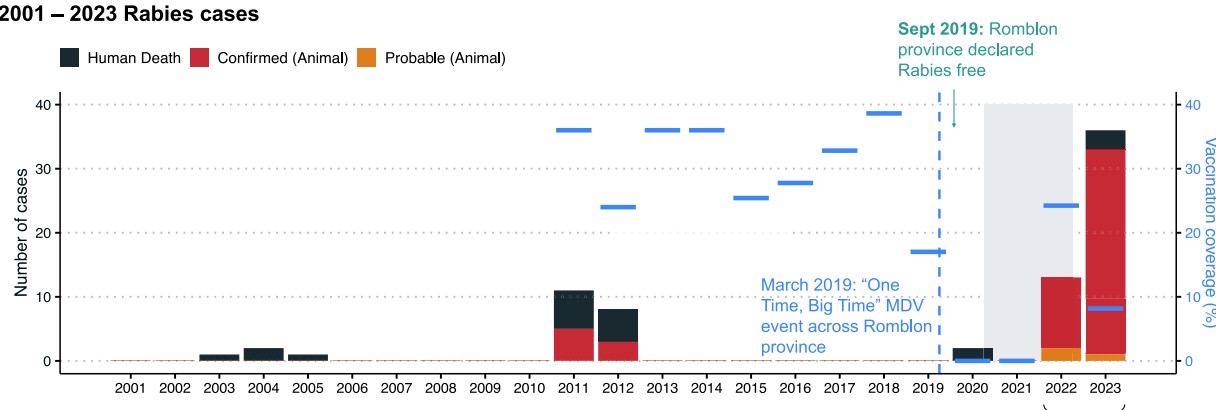

**B    2022 – 2023 Rabies outbreak**

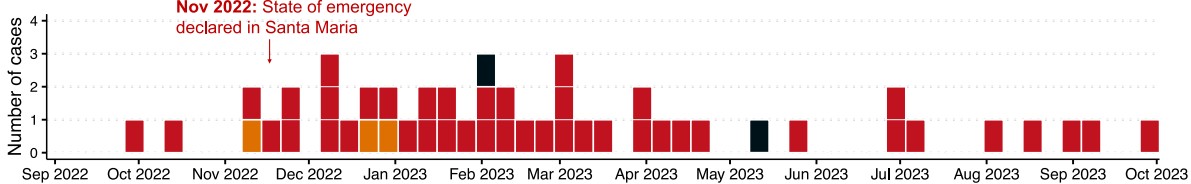

**C**

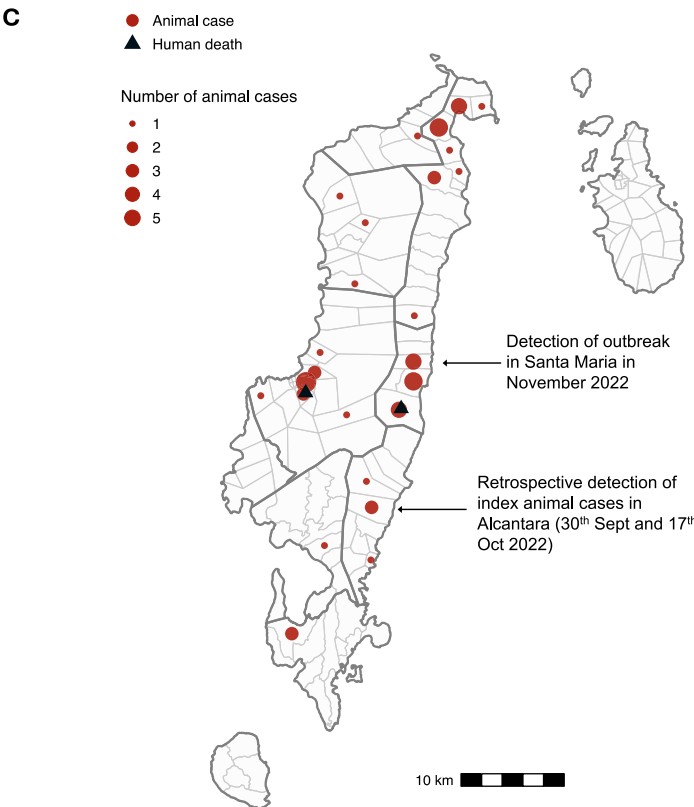

**Fig. 2 | Rabies cases and control measures recorded in Romblon province since 2001. A** Human deaths and confirmed and probable animal cases recorded annually between 2001 and September 2023; this includes two human deaths from Romblon Island in 2020 that were diagnosed based on clinical signs but not confirmed so were not included in official government statistics to retain the province's rabies-free status, and two human deaths in 2023 that were laboratory confirmed following implementation of genomic surveillance. The estimated percentage of dogs vaccinated each year is shown as blue horizontal bars. The shaded area represents the COVID-19 lockdown period. **B** Human deaths, confirmed and probable animal cases recorded during the outbreak between September 2022 and September 2023. Human cases are dated by month of death while animal cases are dated by month of biting incident (if known); otherwise, sample collection date is used (22/43 cases). **C** Animal cases (red circles, scaled by number) and human deaths (black triangles) in Romblon between September 2022 and September 2023. Grey polygons indicate municipalities and cases detected early in the outbreak are annotated.

## Rabies control and prevention

Since 2000, rabies control in Romblon Province has primarily involved yearly mass dog vaccinations, with estimated coverage never exceeding 40% (Fig. 2A). During the COVID-19 pandemic, dog vaccination campaigns were suspended due to social distancing restrictions and resource reallocation leading to a decline in coverage (Fig. 2A). Air travel to and from Tablas Island was suspended from March 2020 to December 2022, but inter-island ferries continued routes via rabies-endemic provinces of Oriental Mindoro, Quezon, and Batangas to ports on Tablas Island in Odiongan, San Agustin, and Calatrava municipalities (Fig. 1C). While pets are allowed on ferries with a health certificate and proof of rabies vaccination, in practice, checks at public ports are rare. Additionally, private pump boats frequently used by fishermen, tourists and visiting families do not subject companion animals to regulatory procedures.

The confirmation of the positive animal rabies case in Santa Maria municipality in November 2022 prompted an immediate state of emergency declaration by the municipal mayor (Fig. 2B). Contact tracing identified humans and animals exposed to the rabid dog, and ring vaccination of 66 dogs within the village was conducted. However, municipality-wide dog vaccination was not carried out due to limited human resources and vaccines. In March 2023, after the first human rabies case, an 'Information, Education and Communication' (IEC) activity, consisting of lectures on rabies prevention, was held in the victim's *barangay* (village). That same month, the governor of Romblon Province instructed all municipalities' mayors on Tablas Island to enforce Republic Act No. 9482 (Anti-Rabies Act of 2007), requiring local government units to allocate funds toward dog vaccination[52]. Subsequent dog vaccinations were both limited and heterogeneous across the island. While some municipalities restarted vaccination campaigns in 2022, at least three did not conduct large-scale dog vaccinations in 2022 or 2023 (Supplementary Fig. S1). As a result the proportion of the dog population vaccinated declined, from 24.2% vaccinated in 2022, to just 8.2% in 2023. A regional workshop held in early 2024 has since catalysed more concerted vaccination that was completed in May 2024 (12,792 dogs vaccinated).

Bite patients are generally administered PEP on presentation to animal bite treatment centres. However, following the release of positive DFAT results, there was an increase in the patients presenting that underwent IBCM risk assessments, with bite victims duly encouraged to complete PEP regimens. Additional contact tracing was conducted but no other bite victims were identified. No human fatalities were attributed to laboratory-confirmed rabies cases. Delays in DFAT results were circumvented, with RDT results used to reinforce tracing of bite victims. However, lack of official recognition of RDTs at national level was seen as an obstacle that prevented regional and provincial managers from declaring cases and a status based on clinical suspicion was presumed to not carry the same weight as the confirmatory test (DFAT). Contact tracing and PEP were also incomplete for some people exposed to one of the probable cases; they had consumed the dog and were hesitant to come forward, fearing repercussions since dog consumption is illegal.

Details of ongoing cases and updates to the epidemiological situation are maintained on the dashboard: https://boydorr.gla.ac.uk/rabies/SPEEDIER/

## Phylogenetic inference

During the outbreak investigation, periodic genomic sequencing was carried out from December 2022 to March 2023. At this time, 96.15% (25/26) of the confirmed positive samples were sequenced. Genome coverage of one out of the 25 sequenced samples was too low for analysis. DFAT confirmed animal brain samples (23/25) had genome coverage of 90–99% while a lower coverage of 88% was achieved for the human skin biopsy sample that was confirmed by nested PCR.

The first sequencing run in December included the first three positive samples from November 2022. The second run on March 2nd, 2023 sequenced the remaining cases from 2022 (including the two earlier cases from Alcantara municipality since sent to RITM), and the first human case. A third run on March 8th, 2023 included the remaining 14 samples up to the most recent at that time (March 1st, 2023) although three prior samples were subsequently traced to the Provincial Veterinary Office. Each run sequenced 12–23 samples, including additional contextual samples from other parts of the Philippines.

A maximum likelihood tree constructed from publicly available sequences from the Philippines ($n = 664$, reducing to 553 after excluding duplicates and consolidating genes from the same sample) plus 28 sequences from this study (24 outbreak cases, 4 isolated from nearby provinces) provided temporal and geographic context for the outbreak sequences. These 581 unique sequences were collected from 1998 to 2023 from different regions in the country and were of varying length (211 to 11,797 bp), covering different regions of the genome (further details provided in the Github repository, see methods). Both time-scaled and substitution-scaled trees for these 581 sequences are shown in Supplementary Fig. S2. Examining the sequences from the current outbreak in the larger contextual phylogeny showed at least three independent introductions to Tablas Island (Supplementary Fig. S2). The largest cluster of cases ($n = 20$) subdivides into three identifiable genetic lineages (1, 4, and 5) based on a patristic distance threshold of 0.0004 (Supplementary Fig. S3 shows a heatmap of patristic distances between the outbreak sequences). These lineages may have been due to either a single introduction or multiple introductions from a single foci over a short period. The second ($n = 1$) and third clusters ($n = 3$) represent another two introductions, each comprising single genetic lineages (2 and 3 respectively, Fig. 3).

The first and largest cluster shares a historical ancestor with sequences from the previous Tablas outbreak (2011–2012). However, ancestral character reconstruction (ACR) determined the earlier and current outbreaks to have emerged from different geographic sources, specifically Pangasinan and Bulacan provinces, respectively, with their time to the most recent common ancestor (tMRCA) estimated as 2010 (Fig. 3). This first cluster shows several polytomies, each with 'star-like' bursts, indicative of an introduction from a common source, succeeded by multiple local transmission chains. The star-like signatures signify rapid dissemination within a naive population[53], making it highly unlikely that these cases resulted from sustained cryptic circulation on Tablas island from the previous 2011 outbreak. Based on the clusters tMRCA, we estimate an introduction around July 2021 (95%CI: Jul 2020–Jun 2022), prior to divergence into three sampled genetic lineages (Fig. 3A). The resulting cases are most closely related to sequences from Central Luzon and National Capital Region i.e. municipalities within Metropolitan Manila, and Bulacan province is the inferred ancestral location (marginal probability of 100%). If the different lineages were from multiple introductions, they all likely arose from Bulacan province.

The second cluster comprised just one case (the sequenced human case with 88% genome coverage), which lies on a distinct outlier branch in the phylogeny (Fig. 3B). It was ancestral to a large cluster of sequences ($n = 167$) from a mixture of geographic regions, including cases from the third cluster (collapsed clade, Fig. 3B), with tMRCA estimated as Oct 1990 (95% CI: Sep 1985–Jul 1994). The geographic source that led to this human case and the time of the lineage introduction, however, could not be pinpointed, likely due to under-sampled diversity in this part of the phylogeny. We estimate that the third cluster resulted from an introduction in late June 2022 (95% CI: Aug 2021–Dec 2022) and was most closely related to sequences from neighbouring provinces within the Calabarzon region, with Batangas province the inferred ancestral location (marginal probability of 97.7%) (Fig. 3C).

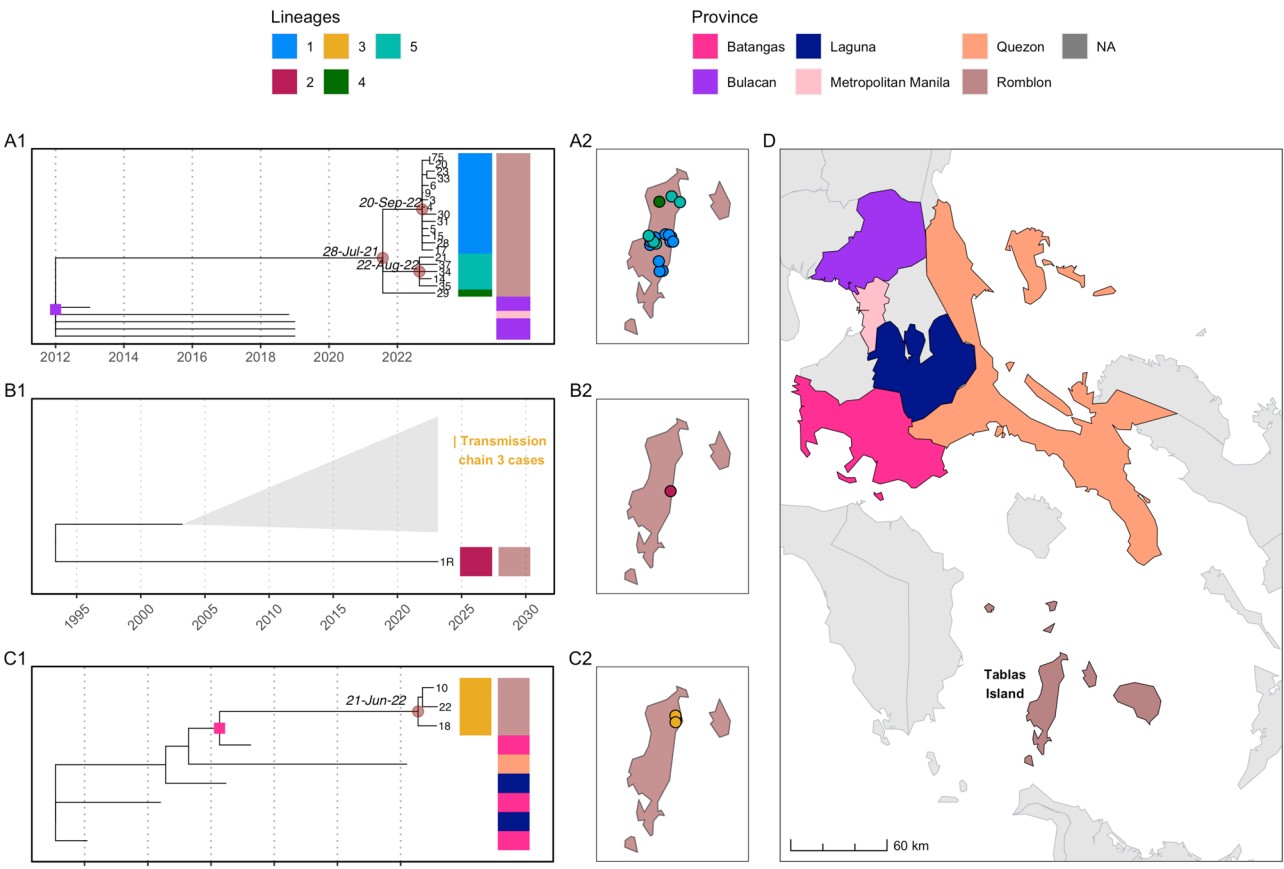

**Fig. 3 | Time-scaled phylogenetic subtrees from the current outbreak.** The source of introductions was inferred by ancestral character reconstruction (ACR). **A1**–**C1** Subtrees corresponding to each inferred introduction, with colourstrips indicating lineages (colours match Fig. 4) identified from clustering by patristic distance (Supplementary Fig. 4) and sequenced case locations (province-level) to match (**D**). Internal nodes mark the tMRCA of each cluster and lineage (circles) and inferred province-level ACR location (squares) for each cluster's ancestral node coloured accordingly. **A2**–**C2** Locations of sequences from Tablas (with points jittered) coloured by lineage and **D** provinces in the Philippines coloured according to ACR.

From simulating a branching process using epidemiological parameters ($R_0$ and serial interval) for rabies viruses, we estimated possible cases resulting from each initial unobserved introduction to Tablas Island. Using the estimated introduction dates for two clusters (1 and 3), we calculated detection delays of 429 days for cluster 1 (from 28/7/2021 to 30/9/2022), and 141 days for cluster 3 (from 28/6/2022 to 14/11/2022). Simulations with realistic incidence suggest a median of 149 undetected cases (95% prediction interval (95%PIE): 14–355) for cluster 1 and 30 (95%PI: 2–180) for cluster 3 before detection. However, if cluster 1 resulted from multiple introductions (estimated around 24/9/2022 and 26/8/2022), leading to lineages 1 and 5, we calculate detection delays of 6 and 121 days respectively, suggesting a median of 1 (95%PI: 1–3) and 20 (95%PI: 1–114.1) undetected cases. Since lineage 4 is represented by a singleton, we presume that it emerged after July 2021.

Transmission trees constructed solely from epidemiological data (dates and locations) were not phylogenetically consistent, highlighting the enhanced resolution provided by viral genomes (Supplementary Fig. S4). Following rewiring for phylogenetic consistency with the five differentiated genetic lineages, pruning by serial interval distribution percentiles (95th, 97.5th, and 99th) resulted in negligible tree configuration changes. Further pruning by distance kernel percentiles led to orphaned cases and short unsampled transmission chains, indicative of either undetected cases in areas with ongoing transmission, or of long-distance human-mediated translocations. Transmission trees inferred using *barangay* centroids versus simulated locations (in proportion to population density) were broadly similar. Using the 99th pruning percentiles split the 5 lineages into 7 transmission chains as indicated by colour in Fig. 4. Lineage 1 split into two transmission chains; lineage 2 linked the biting dog responsible for the first human death with an unsampled dog; lineage 3 remained as one chain and lineage 4 as a singleton; while lineage 5 split into two chains (Fig. 4).

## Discussion

From investigating this outbreak in a formerly rabies-free province, we identified at least three independent introductions that led to rapid island-wide spread. Although 46 animal cases and two confirmed human deaths were detected over the first 12 months, our inference suggests considerable transmission occurred prior to outbreak detection. Long-distance human-mediated translocations and low dog vaccination coverage likely increased the likelihood of both rabies introductions and spread.

Each year, around 200–300 people die of rabies in the Philippines[54]. Mass dog vaccination is effective for rabies control, and has been employed nationwide at varying consistencies. One successful local example is Bohol Province's intersectoral elimination programme, which achieved 70% coverage (as recommended by WHO) through "catch-up" vaccination following mass campaigns[55]. Models predict that vaccinating at least 60–70% of dogs should substantially reduce cases[56], but if coverage is heterogeneous, time to elimination increases, while the probability of elimination decreases[57]. Prior to 2020, vaccination coverage never exceeded 40% in Romblon due to budget and labour shortages, while poor coordination between municipalities led to patchy campaigns that lacked island-wide

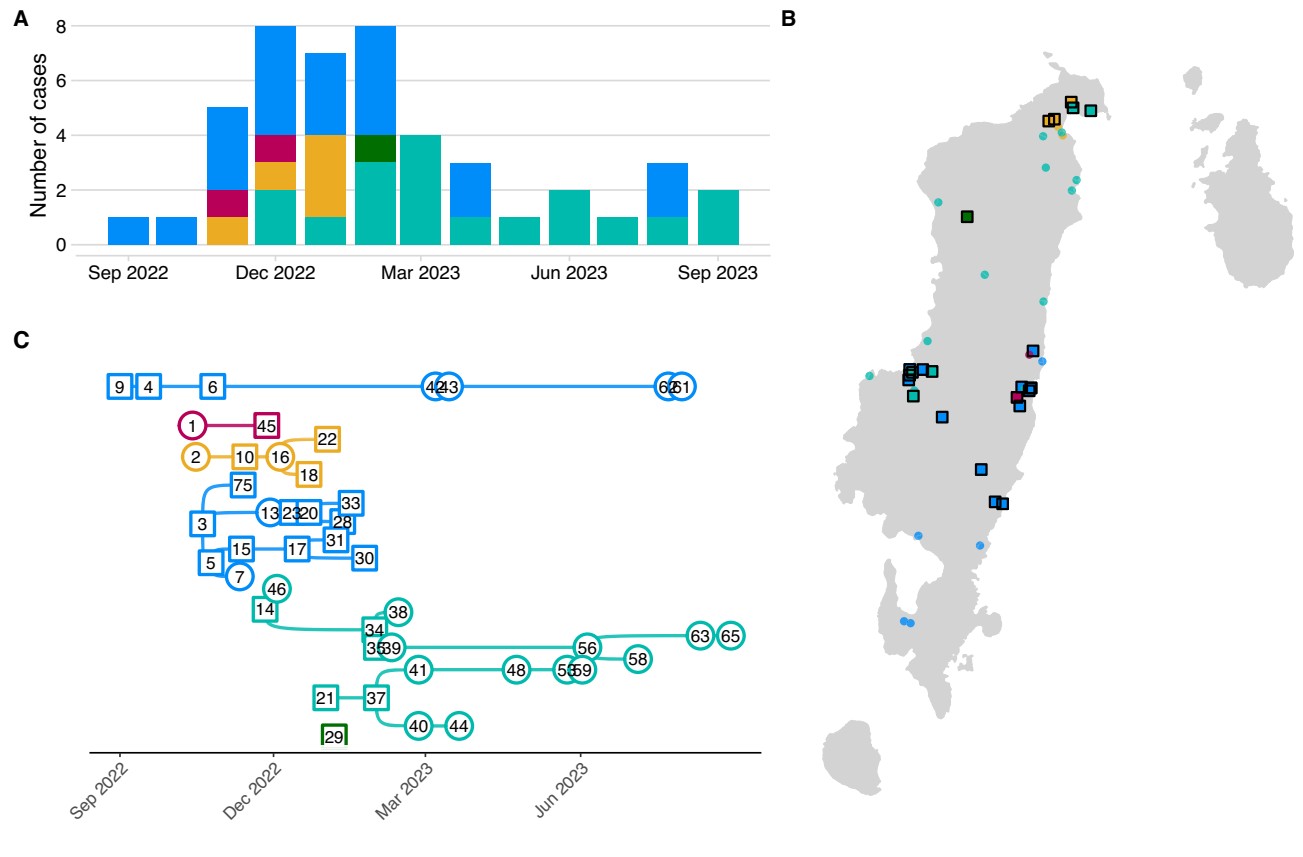

Lineage ■ 1 ■ 2 ■ 3 ■ 4 ■ 5

**Fig. 4 | Inferred transmission chains from the outbreak. A** Monthly confirmed and probable dog cases; **B** mapped dog case locations all coloured by lineage (as per Fig. 3). Squares represent sequenced cases, and circles unsequenced (unsampled) cases, except for case 45, which was not sequenced, but assigned to lineage 2 based on its epidemiological link to the sequenced human case (not shown) and **C** reconstructed transmission chains. The illustrated chains are from the consensus transmission tree with case locations simulated in proportion to human population density and pruning by the 99th percentiles of the serial interval and dispersal kernel. The effects of pruning assumptions and uncertainties on reconstructed transmission chains are shown in Supplementary Fig. S4 and an animation of the consensus transmission tree is provided as Supplementary Movie S1.

coverage. Suspension of vaccinations due to COVID-19 lockdowns in 2020 and 2021 resulted in lower herd immunity across Romblon and much of the country, as evidenced by the subsequent nationwide increase in human rabies deaths, which peaked at 382 in 2023[58] (the highest recorded since 2008).

Our work supports previous findings that incursions occur frequently, with genomic surveillance revealing higher rates than expected[53]. Many introductions fail to take off due to stochasticity in rabies transmission[59]. However, recent outbreaks in other provinces such as Ilocos, as well as the formerly rabies-free island Marinduque, suggest large-scale re-emergence in the aftermath of COVID-19. The third cluster in Romblon was detected in a municipality with a ferry port, possibly linked to an increase in inter-island introductions as travel heightened and restrictions relaxed. The Philippines' archipelagic nature may generally limit incursions in geographically isolated islands, and residual vaccination coverage reduces the chances of secondary cases. However, accessibility to nearby rabies-endemic provinces, coupled with poor vaccination coverage, likely contributed to the outbreak spread.

Prior to the pandemic, gaps in the province's surveillance were apparent: few samples were submitted annually, as storage in the Provincial Veterinary Office and transport by ferry were necessary for confirmation. With staff shortages, timely transfer to the regional laboratory was further impacted by the infrequency of ferry trips due to inclement weather. Apart from causing the suspension of vaccinations, which left dog populations vulnerable, COVID-19 restrictions hindered surveillance, delaying outbreak detection, and possibly leaving earlier outbreaks undetected. Social distancing and prioritised pandemic response impeded investigation of two suspect deaths on Romblon Island in 2020. Samples from animals involved in high-risk bites were mostly not collected prior to the first RDT-positive case, but the result sparked multiple investigations, leading to increased sample collection and RDT use. News of the case result also catalysed testing of two samples that had been stored for over a month. These laboratory-confirmed cases proved that rabies had been circulating earlier than initially presumed.

Between 2017 and 2019, all submitted animal specimens from Romblon tested negative for rabies and no human deaths were reported, reaffirming the province's rabies-free status. Three and zero submissions in 2020 and 2021, respectively, were attributed to lockdown restrictions, while increased case detection and real-time investigation of the outbreak from 2022 onward were enabled by IBCM. Improved communication between animal and human sectors led to identification of most cases, through animal investigations that were triggered by bite victim reports. Use of RDTs may have compelled speedier investigations, since dissemination of positive RDT results increased awareness, which spurred immediate follow up of animals involved in biting incidents[60–63]. Through these timely investigations, animal health workers collected suspicious animals that had died, and the upsurge in testing produced more confirmed cases. To compensate for Romblon's lack of laboratory facilities, multisectoral inter-island collaboration between provinces streamlined the sample

transport process for confirmatory testing and subsequent sequencing. But the volume of samples highlighted challenges, exceeding available resources required to send them individually. Thus, all samples, whether untested, RDT-positive or RDT-negative, were forwarded in weekly batches for laboratory confirmation. When the regional laboratory could not perform DFAT, RDTs were performed instead, before transfer to a third location–the national laboratory (RITM)–for DFAT confirmation and sequencing, requiring additional travel time and exacerbating delays to result reporting.

Sequencing has played a crucial role in informing sources of rabies introductions[25,45,64] and mobilising vaccination responses[24,59]. Integrating genomic data with epidemiological data from IBCM enhanced understanding of the outbreak spread and identified possible points of introduction, also suggesting the need for preventive vaccination, targeting dog vaccination towards source endemic areas. The benefits of genomic surveillance, as evidenced during the COVID-19 pandemic, require that expertise and skills be maintained in-country[65].

Veterinary capacity remains limited across much of Southeast Asia. In Tablas, few staff had to contend with the rapidly evolving public health emergency. The concurrent emergence of ASF prompted government-mandated enhanced surveillance across several provinces, including testing, culling, and banning importation of pork products from affected islands. In comparison, rabies outbreak response was decentralised, fragmented, differing between municipalities, and limited in scale. Prioritisation of ASF by the animal health sector set back investigations, and case confirmation delays slowed the public health response. No formal declaration of an island-wide outbreak was made, and while a few municipalities declared a state of emergency, rabies control efforts were limited. With insufficient dog vaccines and vaccinators, reactive coverage through ring vaccinations of <100 dogs following case confirmation may have provided a small radius of immunity but did little to contain the outbreak, as cases in neighbouring villages showed that transmission was already occurring across shared borders by the time of detection. Moreover, small-scale vaccination is known to be ineffective when only a proportion of cases are detected[56]. Contact tracing and PEP prevented human deaths from confirmed animal cases, but poor awareness constrained contact tracing for probable animal cases, as evidenced by refusal to seek PEP among those involved in consuming dog meat, out of fear of prosecution. The local IEC activity held following a rabies-positive case had limited effects on PEP-seeking and reporting of suspicious animals, as a human death occurred just two months later. Both human cases reportedly resorted to *tandok* instead of PEP, and caretakers only sought medical care after symptoms manifested. Arguably, these deaths could have been averted had communities been sensitised about the rapid rabies spread evident by December 2022. That local response intensified only after a human case, and island-wide dog vaccination was only restarted in 2024, two years after the outbreak's detection, shows how reliance on primarily reactive strategies in health care remains a major One Health challenge.

Our study had several limitations, beginning with IBCM training and support being compromised by COVID-19 restrictions. Despite triggering the outbreak investigation, RDTs were challenging to incorporate into case finding, for several reasons. There was a two-year gap between RDT training and field deployment. Romblon's rabies-free status may have also created a false sense of security, explaining the lack of immediacy in testing suspicious animals. Lack of practice and confidence were reflected in samples that were stored post-collection, with the expectation that the regional laboratory would handle testing. Furthermore, positive RDTs were not considered valid unless matched by DFAT, so there was little incentive to use RDTs as they didn't 'count' as a diagnostic method, even if waiting for laboratory confirmation delayed information dissemination. National authorization for the use of RDTs and release of official diagnostic results could

have expedited early outbreak detection, and if RDTs were recommended internationally, this could perhaps hasten implementation of control measures. Though it must be noted that laboratory confirmation still did not spur outbreak response in several municipalities until a human case was reported.

Genomic surveillance revealed insights not possible from the epidemiological data alone, but were not definitive. For example, the human case sequence points to a second introduction from an unknown source that we were unable to pinpoint, due to under-sampled diversity in the phylogeny. As this sample type (skin) and extraction kit was not ideal (sequencing approaches have been optimised for brain tissue), this sample could be revisited to generate better sequence coverage and depth. Moreover, not all positive samples were sequenced, with only 24/43 early samples sequenced to date. Sequences from samples taken later in the outbreak could reveal which lineages have persisted and if further introductions have occurred. The largest cluster likely resulted from a single introduction from Bulacan province that diverged into three genetic lineages, but it could have resulted from multiple introductions. However, without more sequences from this period and from Bulacan province, we are unable to distinguish these scenarios. Our inference of orphaned cases and short transmission chains indicate either locations with undetected transmission, or long-distance (human-mediated) translocations. Longer delays make our branching process approximation less accurate for estimating undetected transmission, as observed for the largest cluster associated with the initial outbreak detection. Further methodological development could refine these estimates, including accounting for uncertainty in the timing of introductions and for residual vaccination coverage.

Free-roaming dog populations sustain rabies outbreaks worldwide, as seen in Romblon, where most cases were from owned dogs that were unleashed and unsupervised. It is a cultural practice in some LMICs to let dogs wander, and despite local ordinances in the Philippines prohibiting non-leashed dogs in public, these are not easily implemented due to insufficient resources for dog-catching and impounding. Therefore, the burden must also be shared with dog owners to take responsibility for ensuring that their pets are vaccinated and not inconveniencing others.

Achieving vaccination coverage of 70% remains the most important rabies control method. Dog vaccination coverage estimates fluctuated from year to year, ranging between 18–38.6% pre-pandemic and 0–24.2% post-pandemic, hence their limited impact on rabies transmission. Lack of standardised monitoring of coverage meant these estimates were extrapolated from different sources, potentially explaining inconsistencies (Fig. 2A). More generally, heterogeneous coverage in the Philippines is evidenced by lack of coordination among municipalities, even during a deadly outbreak on a small island. If one municipality achieves sufficient coverage, it is still vulnerable to incursions from neighbouring municipalities, highlighting the value of cross-border coordination between local government units. This principle also applies to provinces, as reflected in the increasing number of rabies cases detected post-pandemic (even in provinces lacking IBCM). Henceforth, focus must be placed on proximate control measures in nearby rabies-endemic islands if the 'Zero by 30' goal is to be achieved.

## Conclusion and recommendations

This investigation demonstrates the value of combining epidemiological and genomic data for inferring the source and spread of rabies outbreaks. Enhanced surveillance through IBCM coupled with genomic surveillance proved essential in case-finding and tracking, while simultaneously highlighting the challenges of outbreak detection and response in rural archipelagic settings. The immediacy of RDT results illustrate their potential to inform timely outbreak declaration and

response, but lack of international guidance on their use remains an obstacle.

Despite belying the One Health approach, control measures driven solely by human deaths are unfortunately common in LMICs, with dog rabies cases often not taken seriously. Lessons should be taken from Romblon on RDTs and laboratory-confirmed animal cases acting as triggers for outbreak response. The Philippines has previously demonstrated rabies control capacity, but since its economic impact is negligible compared to ASF (even despite human fatalities), routine surveillance remains neglected and current border control measures at local ports of entry[54] have not been strengthened amidst disease re-emergence. Delayed public health responses that included small-scale ring vaccinations were inadequate, emphasising the need for dog vaccination to be sufficiently large-scale; in this situation, island-wide. Genomic surveillance is beneficial for determining the source of incursions, and can also target preventative vaccination toward rabies-endemic areas. Additionally, sustaining genomic capacity can benefit investigations of other infectious diseases in human and animal populations, with rabies serving as a marker of government response proficiency.

Globally, lessons from this outbreak include proven benefits of the One Health approach in enhancing surveillance, the limitations of short-term control measures, and the importance of routine surveillance in maintaining capacity for responding to potential re-emergence.

## Methods

### Study description
This study took place on Tablas Island, Romblon Province, MIMAROPA region of the Philippines. Tablas has a population of 174,447[66] served by Odiongan, Santa Fe, and Calatrava ports, and Tugdan Airport in Alcantara (Fig. 1). The dog population is not known, but estimates of human:dog ratios in the Philippines suggest it is between 17,445 and 58,149[67]. Prior to 2020, dog vaccination coverage across the province ranged from 18.0–38.6% according to regional reports, and was self-reported by Romblon province from 2021 onwards as fluctuating between 0 and 24.2% (the number of vaccinated dogs per municipality can be seen in Supplementary Fig. S1). IBCM was introduced to Tablas in March 2020, but implementation and in-person training and support was constrained by COVID-19 restrictions enacted mid-way through initial training. Ethical review was secured from RITM ethical review board (2019–2023) and the University of Glasgow, Medical, Veterinary, and Life Sciences ethics committee (200190123).

### Case finding and laboratory confirmation
As part of IBCM, Public Health Workers based in animal bite treatment centres (clinics in hospitals or health units that provide PEP to bite victims) reported 'high-risk' bites to animal health workers at the closest Municipal Agriculture Office (Fig. 5). Bites were classified as 'high-risk' if the biting animal died, was killed, showed signs of poor health, was suspicious for rabies, or disappeared[68]. Animal health workers investigated suspicious animals to confirm their health status. If they required assistance or were busy with other duties, a Disease Surveillance Officer would investigate on their behalf.

Animal investigations were initiated by phone or in-person to gather case details. Sick animals or animals presenting signs of rabies were quarantined at the owner's house, and their health monitored for 10 days[69]. Dead animals were subject to sample collection. Animal health workers retrieved brain tissue, the head, or whole carcass and initiated confirmatory testing in the field using a Bionote RDT[63,70] if available, while packaging an additional sample for laboratory submission. If sample collection did not coincide with scheduled ferry trips, samples were temporarily stored frozen in the Provincial

Veterinary Office, then sent to RADDL 4B for DFAT[71]. As ferry trips were normally scheduled on weekends and outside office hours, one Provincial Veterinary Office staff delivered samples in batches, depending on their availability. When confirmatory testing was not possible at RADDL (fluorescent microscope was broken), samples were instead submitted to the RITM National Reference Laboratory in Manila (Fig. 1). Human samples from virus shedding secretions (saliva) and/or body parts (nuchal skin) were collected through minimally-invasive methods from suspected cases, whether pre- or post-mortem, and sent to RITM for confirmation through nested PCR (Fig. 5)[72]. As part of routine procedures when handling a probable rabies patient, hospitals sought informed consent before conducting sample collection, with a statement clarifying that laboratory results were to be performed for surveillance purposes. Samples that were confirmed positive by DFAT or PCR were stored under cold chain.

### Whole genome sequencing and phylogenetic analysis
Twenty-four out of 43 rabies-positive samples from the outbreak were sequenced, following a previously established protocol for whole-genome sequencing of RABV[73] (Supplementary Table S1) to maximise reagent use, periodic sequencing was conducted, with 12–23 samples per run.

A Romblon-only phylogenetic tree was generated in IQtree and Romblon sequences were divided into phylogenetic lineages, for transmission tree inference[74] (see next section). Lineages were defined through patristic distance clustering with the adegenet package[75] using a threshold of 0.0004, determined by comparing patristic distance clusters with phylogenetic trees and considering the RABV evolutionary rate ($-2 \times 10^{-4}$ substitutions/site/year). A heatmap of patristic distances is available in Supplementary Fig. S3.

A large contextual dataset of partial and whole genome Philippine RABV sequences ($n = 615$) from the RABV-GLUE database ($n = 694$)[76] was obtained for this study, with additional recently published whole genome sequences (WGS)[77] ($n = 49$), and WGS from an ongoing Philippine RABV study ($n = 4$) and this outbreak ($n = 24$). The contextual data constituted RABV sequences with the Philippines as the country of origin, which belong exclusively to the Asian SEA4 clade, a phylogenetic clade associated with and almost entirely restricted to the Philippines (we did not include 11 sequences found in other countries). Associated metadata was prepared using custom R scripts to clean and standardise data from the different sources, including merging of sequences from different genes with the same isolate ID. Overall, this resulted in a dataset of 581 sequences. Metadata, sequences and code can be found in the GitHub repository. To prepare a sequence alignment, concatenated WGS ($n = 79$) were aligned using MAFFT (v7.520)[78] with default parameters, then added to the RABV-GLUE downloaded alignment using MAFFT's functionality to add full-length sequences to an existing multiple sequence alignment with the keeplength option on. Each alignment was checked and edited manually as required (minor edits). Partial sequences with the same sample IDs (but submitted under different GenBank accession IDs) were merged using another custom R script.

Phylogenetic analysis including tree dating and ancestral character reconstruction was performed following the methods in ref. 79. A maximum likelihood tree was constructed from the 581 Philippines RABV sequence data using FastTree v2.1.11[80] with a GTR+Gamma20 model. Using the sequence-associated dates, this tree was rooted according to the best root-to-tip correlation, by running the initRoot function in R package BactDating (https://github.com/xavierdidelot/BactDating). The rooted tree was pruned to WGS only using gotree (v0.4.5) and the evolutionary rate estimated using the R-wrapper for lsd2[81] (Rlsd2) with a ZscoreOutlier of 3. This rate estimate was used as a prior to inform tree dating for the full 581 sequence RABV tree with a

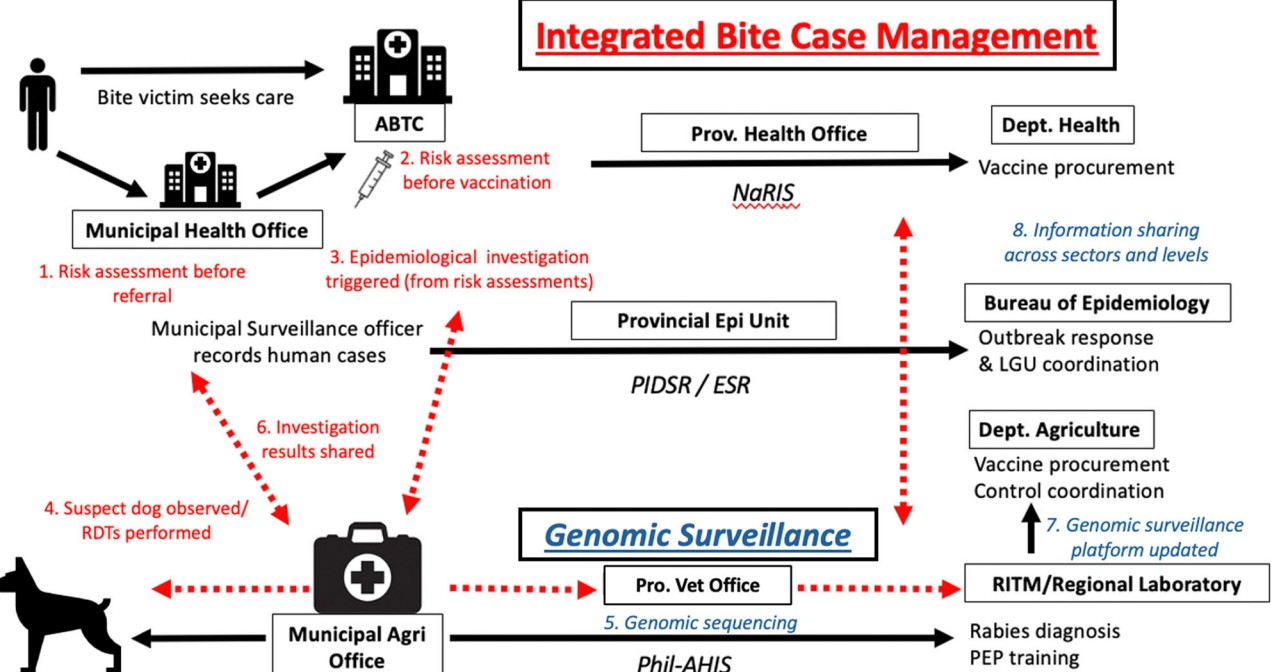

**Fig. 5 | Diagram of the complete integrated bite case management (IBCM) process.** Activities that enhance surveillance through IBCM shown in red, while genomic surveillance activities are shown in blue. ABTC Animal Bite Treatment Centre, LGU Local Government Unit, NaRIS National Rabies Information System, Phil-AHIS Philippine Animal Health Information System, PEP Post-exposure prophylaxis, PIDSR/ESR Philippines Integrated Disease Surveillance and Response/ Event-based Surveillance and Response, RDT Rapid diagnostic test, RITM Research Institute for Tropical Medicine.

ZscoreOutlier of 5 and 1000 bootstraps to generate date CIs. PastML[82] (v1.9.43) was used to perform ancestral character reconstructions on the dated tree using a marginal posterior probabilities approximation, with Philippines administrative divisions as traits (region and province). Subtrees including recent Romblon outbreak sequences and their 10 closest relatives were extracted from the larger contextual phylogeny for interpretation. Romblon phylogenetic lineages were defined according to a patristic distance threshold of 0.0004. Tree annotation and visualisation was performed in R with the ggtree package[83].

Outbreak spread between introduction ($T_{int}$) and first detection ($T_{obs}$) was estimated using a branching process model, simulating the serial intervals and secondary cases probabilistically from lognormal (meanlog = 2.85, sdlog = 0.966) and negative binomial distributions (mean = 1.20, $k$ = 1.33), respectively, to generate descendents from the initial case[84,85]. The interval between $T_{int}$, inferred via phylogenetic analysis, and $T_{obs}$ was calculated (not accounting for the tMRCA uncertainty) to determine simulation run time, conditioned on outbreak persistence until $T_{obs}$. 1000 outbreaks were simulated, and the median and 95% prediction intervals of undetected cases calculated. As this model assumes an infinite susceptible population, the median and prediction interval were calculated only from plausible outbreaks (incidence not exceeding 1% of Romblon's dog population).

## Transmission Tree Inference
We probabilistically reconstructed transmission trees using the treerabid R package v1.0.1 that generates trees consistent with phylogenies[86]. Progenitors for each case were inferred from reference distributions of the rabies dispersal kernel and serial interval (Lognormal serial interval, meanlog 2.85, sdlog 0.966, and Weibull distance kernel, shape 0.698, scale 1263.461)[84]. We incorporated uncertainties into our bootstrapping procedure for dates of case onset and case locations, since the *barangay* was recorded for each case but geolocations were not. Specifically, for each bootstrap, we assigned case

onset dates by sampling uniformly from a 5-day window up to and including the date of the biting incident or sampling if this was reported, or a 15-day window up to and including the date of laboratory submission or testing otherwise. We selected plausible case localities by sampling from 100 × 100 m raster grid cells in proportion to population density according to unconstrained model data from worldpop[87].

We generated 1000 bootstrapped trees for each of 32 scenarios, corresponding to all combinations of the following: (i) case locations (*barangay* centroids versus locations sampled from the population density grid); (ii) use of genetic data for inference (yes/no); and (iii) inclusion of pruning steps to further resolve transmission chains (eight different combinations of cut-offs). In the scenarios using genetic data, transmission trees were first constructed using spatiotemporal data as described above, and then made consistent with phylogenetic lineage assignments by using a rewiring algorithm for cases assigned to incongruent lineages. In the phylogenetic lineage assignments, we interpolated the existence of an unsampled rabid dog at the time and location of the human exposure that developed rabies and for which a sequence was obtained, and assumed this case belonged to the designated lineage. In the scenarios with additional pruning steps, case pairs were filtered out where the time interval and/or distance exceeded specified percentiles of the serial interval and distance kernel distributions. Pruning options included no pruning, pruning by time only (cut-offs 0.95, 0.975, and 0.99), pruning by time and distance using the same cut-offs (0.95, 0.975, and 0.99) and one combination of differing cut-offs (time 0.975, distance 0.99). Without pruning or integration of phylogenetic information, tree reconstruction results in a single chain. The different scenarios were compared on the basis of their consensus trees.

## Reporting summary
Further information on research design is available in the Nature Portfolio Reporting Summary linked to this article.

## Data availability

The rabies surveillance and phylogenetic data used in this study are available in the Github repository:https://github.com/boydorr/outbreak_romblon_PHL. The genome sequence data are provided in Supplementary Table S1, while the GenBank accession numbers are provided in Supplementary Table S2.

## Code availability

The code used in this study, and all additional licenses and copyright notices are provided in the Github repository: https://github.com/boydorr/outbreak_romblon_PHL[88-90].

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

## Acknowledgements

We are grateful to all the Romblon public and animal health staff who have been working hard to support the outbreak response, including Mitch Famaran, Victor David, Cliff Richard Mabasa, and Ariel Malicse. We appreciate the administrative support provided by FETPAFI and the SPEEDIER (Surveillance Integrating Phylogenetics and Epidemiology for Elimination of Disease: Evaluation of Rabies Control in the Philippines) team. This work was funded by grant [224520/Z/21/Z] from the Wellcome Trust, Newton funding from the Medical Research Council [MR/R025649/1] and the Philippines Department of Science and Technology (DOST), a DOST British Council-Philippines studentship (C.B.), the UK Medical Research Council (KB)[MR/X002047/1] and an EPSRC studentship (R.D.)[EP/T517896/1].

## Author contributions

K.H., M.Y., C.T.B., K.B., and M.E.G.M. contributed to conceptualisation (ideas; formulation or evolution of overarching research goals and aims). C.T.B., E.R., K.B., V.D.D.C., M.Y., and C.B. contributed to data curation (Management activities to annotate/produce metadata, scrub data, and maintain research data including software code, where it is necessary for interpreting the data itself, for initial use and later reuse). C.T.B., K.B., E.R., M.Y., R.D., and C.B. contributed to formal analysis (application of statistical, mathematical, computational, or other formal techniques to analyze or synthesise study data). K.H., K.B., C.T.B., C.C., M.E.G.M., and N.R.C. contributed to funding acquisition (acquisition of the financial support for the project leading to this publication). K.B., C.T.B., M.Y., V.D.D.C., J.M., D.L.M., D.R.M., A.F., L.N., H.A., J.K.B., J.B., E.P., S.V.M.T., and J.M. contributed to investigation (conducting a research and investigation process, specifically performing the experiments, or data/ evidence collection). K.B., C.T.B., K.H., R.D., and E.R. contributed to methodology (development or design of methodology; creation of models). M.E.G.M., K.H., K.B., and N.R.C. contributed to project administration (management and coordination responsibility for the research activity planning and execution). Carlijn B., E.R., K.B., and M.K. contributed to software (programming, software development; designing computer programmes; implementation of the computer code and supporting algorithms; testing of existing code components). K.H., K.B., M.E.G.M., C.C., N.R.C., M.Y., and D.M. contributed to supervision (oversight and leadership responsibility for the research activity planning and execution, including mentorship external to the core team). E.R., C.B., K.B., and C.T.B. contributed to visualisation (preparation, creation and/or presentation of the published work, specifically visualisation/data presentation). M.Y., C.T.B., and V.D.D.C. contributed to writing—original draft preparation (creation and/or presentation of the published work, specifically writing the initial draft, including substantive translation). K.H., E.R., C.B., M.E.G.M., N.R.C., and K.B. contributed to writing—review & editing (preparation, creation and/or presentation of the published work by those from the original research group, specifically critical review, commentary or revision—including pre- or post-publication stages).

## Competing interests

The authors declare no competing interests.

## Additional information

[1]Boyd Orr Centre for Population and Ecosystem Health, School of Biodiversity, One Health & Veterinary Medicine, College of Medical, Veterinary & Life Sciences, University of Glasgow, Glasgow, UK. [2]Field Epidemiology Training Programme Alumni Foundation Inc (FETPAFI), Quezon City, Philippines. [3]Research Institute for Tropical Medicine (RITM), Alabang Muntinlupa City, Metro Manila, Philippines. [4]School of Computing Science, College of Science & Engineering, University of Glasgow, Glasgow, UK. [5]School of Mathematics & Statistics, College of Science & Engineering, University of Glasgow, Glasgow, UK. [6]Regional Animal Disease Diagnostic Laboratory, Naujan, Oriental Mindoro, Philippines. [7]Municipal Health Office, Alcantara, Romblon, Philippines. [8]These authors contributed equally: Mirava Yuson, Criselda T. Bautista, Kirstyn Brunker, Katie Hampson. ✉e-mail: m.yuson.1@research.gla.ac.uk

