## [Transparent Peer Review file · Nature Communications]

Combining genomics and epidemiology to investigate a zoonotic outbreak of rabies in Romblon Province, Philippines

Corresponding Author: Dr Mirava Yuson

Version 0:

Reviewer comments:

Reviewer #1

(Remarks to the Author)

The article entitled “combining genomics and epidemiology to investigate a zoonotic outbreak: rabies in Romblon Province, Philippines” provides a retrospective description of a rabies outbreak in a Philippine archipelago, that was considered dog rabies free for less than a lustrum, using classical descriptive epidemiology methods combined with phylogenetic analyses. Authors described the temporal spatial distribution of rabies cases during the recent outbreak 2022-2023 (using 43 positive animal cases and 2 humans), which were placed in context with historical information on rabies cases in animals and humans, animal vaccination coverage gathered since 2003 and 2011, respectively (figure 1). Authors used an existing integrated bite case management system couple to laboratory-based rabies diagnosis as a sentinel proxy for the early detection of potential rabies outbreaks in the Philippines. To have a more detailed insight on how rabies could have appeared in Romblon province, authors used almost complete rabies virus genomic sequences for 23 animals (with a genome coverage ranging from 90 to 99%) and 1 human (with 88% genomic coverage), which were put together with a contextual historical and contemporary data set of 746 unique sequences obtained from 1998 to 2023. In general, authors used sound and recent methodologies for both epidemiological and phylogenetic analyses, which allow them to infer the rabies outbreak studied was likely a consequence of at least three independent introductions from multiple ongoing rabies foci across the Philippines. Highlighting, how well interconnect is this topographically highly complex archipelago. The phylogenetic reconstructions analyses reveal several potential gaps in conventional surveillance systems. One of the most important ones is a detection delay of rabies cases in the main reservoir host (in this case dogs), or in vectors (cats) inherent to the detection threshold inflicted by a lack of a 100% surveillance coverage across the urbanization gradient (encompassing highly populated areas to marginalized or isolated regions), the overall public awareness on rabies, and limited resources allocated for laboratory-based disease detection, and use of less sensitive “more affordable” laboratory-based diagnostic methods. The discussion is compelling and complete, including all serious gaps within the overall investigation. The conclusions reflect most of the findings although some of them are ambiguously generic public health statements not necessarily based on the results of this investigation.

Suggestions and recommendations.

Introduction section

Lines 57 and 90. In line 57 authors refer dogs as the primary vector for rabies, being responsible for the transmission of rabies to humans. However, in line 90 mentioned the use of genomic surveillance to decipher reservoir dynamics. In general, the term vector is used to refer an intermediary necessary to maintain the life cycle or transmission cycle of a parasite (from viruses, protozoa or helminths). However, the most common term used across the rabies field is reservoir host, since different rabies virus (RABV) clades or variants are maintained in an independent fashion by specific species of mesocarnivores worldwide or by different bats species (only in the Western Hemisphere). Dogs are the primary and most widespread reservoir host of several distinctive major and minor RABV clades, which depict circumscribed/unique geographic distributions across the globe. Only in very rare instances, dogs and cats may work as vectors of RABV variants established in other major rabies reservoir hosts such as vampire bats. Thus, authors should harmonize their terminology referring to dogs as a major rabies reservoir host and not as a vector, since that is what dogs are in the Philippines major rabies reservoir hosts and not vectors.

Results section

Phylogenetic inference subsection.

For this study authors produced nearly complete genomes for 24 samples, encompassing 23 rabid animals and 1 human, using 746 unique sequences with different nucleotide lengths (from 211 to 11797 bp) as an epidemiological contemporary

and retrospective (historical) context to decipher RABV dissemination dynamics across the Philippines. However, authors do not mention anything about these contextual sequences. It is not clear whether all these contextual sequences only came from the Philippine, or there may be other sequences from neighboring countries. Thus, I consider pertinent authors should include a detail table indicating GenBank accession number, year, geographic location across the Philippines, species of each of these sequences, designated lineage, sequence length and reference if previously published. I do recommend a table using the same color codes for provinces as indicated in figure 2.

Lineage definition

There is no clarity in the criteria used for the definition of lineages, as well as the extraction of subtrees (Figure 2) oversimplifies the original topological characteristics better shown in the original trees.

In the case of lineage definition, authors mentioned they considered patristic distances as their main criteria for lineage definition, however there is no clear distance threshold they showed that delineates the inter-lineage and intra-lineage variation ranges. Authors should show a heat patristic distance matrix showing the average inter-lineages patristic distances, so readers can have a sense of the extent of genetic variation across lineages.

In regards lineage definition most authors first define statistically supported clades and subclades (high bootstrap values or posterior probabilities) as putative lineages across the complete trees, to later check if there is a correlation among sequences pertaining to specific clades and the geographic location they were obtained from. Lineages are usually color coded from the region or country where most of the sequences pertaining to that clade come from. In this study lineages were color coded differently, given the impression there is no connection between lineages and the geographic location they came from. It would be valuable for potential readers to see if there is a correlation among between clades and the geographic region across Philippines they came from. Ancestral state determinations would corroborate such visual inferences.

For example, line 224, Authors refer to a “star-like phylogeny” or tree topology, when what is mainly depicted across Fig 2 subtrees looks more like polytomies in subtree 2 A1, while subtrees 2B1 and 2C1 may show a more star like topology. I think a star-like topology would be best appreciated the original trees shown in the supplementary figures if they are color coded appropriately.

Reviewer #2

(Remarks to the Author)

This paper reports the detection and response to a rabies outbreak on an island province in the Philippines, making use of modern whole genome sequencing technology to gain insights into rabies introduction events, viral transmission, and rates of case detection. The study demonstrates the fundamental importance of strengthening One Health surveillance processes to progress national rabies control initiatives, as well as the global agenda for the elimination of dog-mediated human rabies by 2030 (Zero by 30). The experience reported in the use of rapid field-side tests is of benefit to increasing the detection of rabies outbreaks and evaluating rabies burden at scale in resource limited settings.

The study presents a case for the enhancement of rabies surveillance through the use of modern gene sequencing technology, however more context is required around the feasibility of establishing this capacity as a routine component of rabies surveillance in most endemic settings. The paper emphasizes the benefit that real-time genomic surveillance brings to the effective design and implementation of rabies control interventions, and uses the comparison of Ebola, Lassa Fever and Mpox (Line 87). It would be pertinent to discuss the operational methods and challenges of implementing whole genome sequencing in the study site and establishing real-time genomic surveillance more broadly. For example, it would be helpful to report in the methods how often sequencing was performed and whether it has been possible to incorporate sequencing as an ongoing component of the rabies control program in the Philippines. It would be beneficial to report the average time between laboratory rabies diagnosis and identification of virus lineage in this study, as this is of relevance for WGS to be used as a tool in guiding outbreak response. If routine in-country genomic sequencing is yet to be established, it would be important to discuss why this is the case and whether widespread use of routine genomic surveillance is likely to be feasible in most rabies endemic settings within the Zero by 30 timeframe. Practical guidance on when genetic sequencing is most useful and possible ways of maximizing its benefit in situations where only a fraction of samples can be sequenced would be helpful.

The authors mention the currently inadequate capacity for field-investigation/vaccination and laboratory diagnosis of all submitted cases. Therefore it would be important to frame whether it would be more pertinent to direct limited funds towards establishing widespread investigation and testing of suspect rabid animals, before prioritizing the development of viral sequencing, or whether there would be possible scenarios where advancement of both could be feasible. These constraints are significant when advocating priorities for advancing towards Zero by 30 and in enabling stakeholders in rabies control elsewhere to consider how genomic sequencing may be of relevance in their setting.

Greater description of the relationship between insights from gene sequencing and vaccination activities would help to showcase the working impact of the tools described. Whilst the discussion mentions the need to improve the initiation of a timely, coordinated response following the detection of rabies cases, the significance of genomic sequencing on guiding field strategy and vaccination response in this outbreak was not clear. It appears that municipal-level data on vaccination output was not available, but some information on the scale and distribution of vaccination effort is reported in the discussion (Line 327). Some information is given about vaccination data availability in the caption of Figure 1 (Line 197 – 198), however this seems misplaced and availability of vaccination data across the study period should be included in the methods. The results would be improved if it were possible to include data on where and when vaccinations took place on Tablas during

the study period in relation to local rabies case incidence. Transmission of lineages 2, 3, and to some extent lineage 1, appear to have only sustained from Oct 2022 to Feb 2023, however there is no mention of why these groups then appeared to disappear by March 2023 (Figure 3). Did a vaccination campaign take place towards the end of 2022 in the regions of Tablas where these groups circulated, or was it more likely that these lineages became extinct in the absence of a recent campaign? The study by Bourhy 2016 (DOI: 10.1371/journal.ppat.1005525) reporting natural extinction of rabies virus introductions in CAR and is probably more relevant than the Ebola reference used in Line 308. The need for improved quality and access to data on dog vaccinations would seem to be an important learning from this outbreak to better coordinate the vaccination response.

The clear presentation of spatial information to support outbreak response and campaign planning is core to the premise of the paper, however variation in the formatting of cartographic figures is currently disorientating. Keeping styling and extent of the Tablas maps consistent between Figure 1(D), Figure 2 (A2/B2/C2) and Fig 3(C) would aid the reader in orienting the maps between figures. Figure 1C provides useful context to the study site (Tablas island) in relation to surrounding geography, but this comes after surveillance data are being presented in parts A, and B, so the reader has to jump around a bit to work out where cases are occurring. An initial figure providing the key geographic locations and boundaries would be a useful reference for understanding the spatial context of data presented throughout the paper. Perhaps Figure 1C could be pulled out as its own figure with labels for Romblon province and the text labels for RADDL, RITM and Manila made clearer. Santa Maria is referred to repeatedly in the paper, but this is not explicitly labelled anywhere in the figures, so perhaps a reference map of Tablas island as part of this figure also showing the key ports and other locations referenced in the results (Odiongán, San Agustín etc) would aid interpretation of the surveillance data.

Specific comments:

Line 61 – The figure of 161 daily global rabies deaths appears to be calculated from an estimate from almost ten years ago and there have since been lower estimates (e.g. Gan et al 2022, 10.1016/j.ijid.2022.10.046). Please update the statement to more accurately reflect the reference and that this was an estimate.

Line 75 – The reference used to support the statement about incursions / importations across borders occurring regularly worldwide is a study focused on a single city (Arequipa). Please add references to support the worldwide nature of the comment.

Line 96 – Please check the sentence for correctness “...detection of a rabid dog last 2022 on the island of Tablas...”. Is the part “...last 2022...” a typographical error – should it read “...in 2022...”?

Line 290 – Is there a source reference for the annual rabies deaths in the Philippines?

Line 293 – It would be relevant to not only state the recommended target annual vaccination coverage to control rabies, but also to emphasize the importance of homogenous coverage across contiguous populations to realize elimination. It is mentioned earlier in the paper that vaccination activity was not equal across all municipalities, which could significantly undermine the likelihood of achieving elimination (Ferguson 2015 (DOI: 10.1038/srep18232), Townsend 2013 (DOI: 10.1371/journal.pntd.0002372), Suseno 2019 (DOI: 10.20506/rst.38.1.2954)).

Line 347 – It seems of limited benefit to advocate for enforcement of dog leashing in a locality which has neither sufficient infrastructure nor enforcement capacity, and also where this practice is counter to cultural norms. Perhaps include a greater emphasis on the need for owners to take responsibility for the public health impact of dogs in their care, by ensuring that dogs are vaccinated and do not cause a public nuisance.

Line 375 – Is use of the word ‘parsimonious’ appropriate in this context? The definition of parsimonious describes someone unwilling to spend money, which is confusing when applied to this sentence.

Figure comments:

Figure 2 – This figure shows a lot of useful information, however is currently challenging to understand across all the components. The positioning of the fill color legend for lineage above the phylogenetic trees (A1 – C1) and for province above the map (D), make it initially difficult to understand that the first color strip in the tree figures relates to the Province fill legend above the map on the other side of the figure when reading from left-to-right. It might be clearer to either put the province fill legend first in the top left, followed by the lineages legend, but perhaps with the legend title as “Romblon lineages” to make it clear that these are the cases from Romblon. It would be helpful to add a reference point to the province map to help orient where Tablas is. The Tablas maps could be cropped to match the extent in Figure 1(D), which would enlarge the island to show the locations more clearly. The color of Romblon province in parts A2-C2 appears to be a lighter shade of that in the province map, however this introduces another color that does not match any of the other labelled colors. You could change Romblon in the province map to the lighter color for consistency throughout the figure.

Figure 3 – I’d suggest putting the legend fill color label in the top left corner of the figure so that it is the first thing seen when reading for the first time left-to-right. Sub-labels of A / B / C are not currently shown within the figure, but are referenced in the caption. It is slightly confusing that colors are conserved between Figure 2 and 3, but icon shape convention is not consistent – in Figure 2 circles are sequenced cases and squares are inferred province-level ACR location, whereas in

Figure 3 squares are sequenced cases and circles are inferred lineage of un-sequenced cases. Could the squares / circles be reversed in one of the figures to help with continuity in designation of sequenced cases across the figures?

Reviewer #3

(Remarks to the Author)

General comments

Yuson et al. present an interesting paper describing the problems encountered to investigate and mitigate a zoonotic outbreak of rabies in Romblon Province, Philippines. They suggest that combining rapid whole genome sequencing (WGS) and epidemiology of rabies viruses (RABV), one of the major neglected tropical disease causing a large public health and economic burden in Africa could reduce the lag time from sample to epidemiological action in LMICs and improve the implementation of control measures by providing a better and more rapid understanding of disease transmission dynamics. The article is well written, the findings are important for guiding rabies surveillance and control measures in the Philippines, and the study is a nice illustration of the importance of combining IBCM and genomic surveillance for rabies outbreak investigations and further demonstrate the importance of prompt implementation of control measures to limit the spread of rabies after re-introduction in a rabies-free area. However, the article is written in a way that gives the impression that the approach is novel when in reality several other rabies studies have used similar approaches, which were not referenced in this manuscript.

The concept of this paper has already been presented in a previous paper co-signed by some of the same authors and showing how the WGS and the management of genomic data could play a role in a surveillance context in a different country (see Lushasi et al., *Elife*. 2023 May 25;12:e85262).

In this study, the authors claimed that they demonstrate the interest of real-time sequencing of RABV to rapidly inform policy decisions and disease management. However, this study neither demonstrates the feasibility of real-time sequencing of RABV in the setting of Romblon province nor does it demonstrate its impact to inform policy decision and disease management. This demonstration is impaired by many limitations (important delays between bite and laboratory diagnosis, difficulties in shipping the samples, lack of confidence in the processes, set up of new procedures after the re-introduction) that are presented in the discussion. These limitations dramatically weaken the ability of this approach to accurately inform epidemiological surveillance and complex epidemiological questions in real time and this study does not bring any significant clues to solve them.

Further, some of the methodologies need to be described in more detail (such as the IBCM approach and the simulation work). Finally, most of the conclusions are not supported by the data presented (see below comments on Line 393-397). In general, this is an important article but does not meet the novelty standards of a large audience journal.

Specific comments:

Line 29-30: suggest mentioning other aspects of the global strategy to eliminate human deaths such as improving surveillance and community participation.

Line 35-36: why put so much emphasis at this stage on the role of suspension of routine dog vaccination due to COVID-19 restrictions. This is not demonstrated in the paper.

Line 45-55: Suggest discussing rabies in the context of neglected tropical diseases at the start of the introduction rather than emerging infectious diseases. It could be put in the context of EIDs later on in the introduction when discussing re-introductions.

Line 47-49: There are many examples of re-introduction of rabies in areas where it was previously eliminated. They should be cited.

Line 50-53: Rabies also requires surveillance and community awareness campaigns, not just PPE and mass dog vaccination. Suggest also adding references here.

Line 53-55: Suggest being more specific. The current arguments are quite broad making it difficult to really see what the barriers are.

Line 57-58: Primarily in Africa and Asia

Line 61-62: Here it is worldwide but in the abstract it's only in LMICs.

Line 66-67: Could the authors provide any evidence of the role of dog vaccination in the elimination of dog rabies in Europe in the beginning of the 20th century. There may be a confusion here with the elimination of fox rabies at the end of the 20th century.

Line 69-72: It's difficult to follow the reasoning for the examples selected. Suggest citing other well-known studies in African settings, in Chad for example.

Line 73-79: It seems that there is a confusion here between mass vaccination of dogs to eliminate rabies from an enzootic area and to prevent re-introduction into a free area. Could the authors provide examples on how these strategies may differ in terms of % of dog vaccination coverage in various settings?

Line 82-87: Many tools and approaches are mentioned without giving any reference to rabies although there are many regarding the use of RDTs, mobile communication and sequencing. Instead there are many citations regarding other zoonotic diseases.

Line 85: Suggest defining "real-time" since the methodologies indicate multiple day delays until diagnostic testing and sequencing, suggesting that the approach is not real-time per say.

Line 85-91: Unclear why the others present genomic surveillance using other zoonotic diseases as examples when there is quite a lot of literature on rabies genomic surveillance. Suggest prioritizing the rabies literature.

Line 130: At this stage, provide references on the use of RDTs for rabies diagnosis rather than the reference of the kit which can be given in M&M.

Line 137-139: These delays are long. Could the authors provide the delay between death of the animal and diagnosis? It

should be around 14-15 days if I am not mistaken. What could explain such a long time? How could this rapidly inform therapeutic decision? What was the delay between exposures and PEP in the treated patients? How could the whole surveillance system be reactive and sequencing data be informative if such delays remain?

Line 141-145: These data are not very specific of this study and not very informative regarding the objective of the study. I suggest to present them in a supplementary table.

Line 145: How does % of rabies positive dogs reported as previously vaccinated compare to % of rabies-negative dogs? Is it significantly different? How does this compare with the estimation of % of coverage given for the province?

Line 170: Does the confirmation occur in November regarding the delay of diagnosis (date of case + 23 days)?

Line 199 (Fig 1): What do the dates provided correspond to: date of death, date of diagnosis?

Line 207: 23 DFAT positive samples are mentioned but 43 were tested positive by RTD. Explain the difference?

Line 211-212: Need to add information on sample sizes by year and geographical location. Same comment for sequence length and regions of the genome that were sequenced.

Figure 2: Suggest adding the full tree as a first panel to show the 3 clusters (as opposed to putting it in supplementary materials)

Line 223-237: It appears that there is lot of uncertainty regarding the number of importations given by the phylogenetic analysis and considering the fact that only 24 samples out of 43 positive ones were sequenced. This is expected and it is known that such a study could definitely underestimate this figure. The authors should then be more cautious. There are probably many more introductions than those reported by the phylogenetic analysis. This is one of the major limitations of the phylogenetic approach. For an illustration, see a study performed in Bangui in Central African Republic which clearly shows how underreporting, limitations in sample collection and the rate of evolution of the RABV genome impair the resolution of introduction events and the resolution of transmission chains based on phylogenetic studies compared to what can be evaluated by modeling (Bourhy et al., PLoS Pathog. 2016 Apr 8;12(4):e1005525). Therefore, the detection delays calculated line 241 should also be given with caution.

Line 290: Need a reference here.

Line 292: May the authors comment the present epidemiological situation of Bali regarding rabies despite mass dog vaccination. How does this compare with Bohol?

Line 305 "Our work and that of others": from a general point of view there are many interesting and important studies from other groups that are not mentioned. Suggest that the authors could make some efforts to mention other instrumental studies and reviews in that field.

Line 308: Again, there are studies dealing with RABV illustrating this concept. No need to refer to EBOLA (REF #32).

Line 314-319: Data on submissions between 2011 and 2022 are not available. Please document. Illustrate the fact that the case detection rate increased from 2022.

Line 386: Could the authors provide data showing "increased case detection due to IBCM promoted communication between animal and human sectors across several provinces". This statement is so far not supported.

Line 393-397. The authors stated that their study demonstrates 4 different points. The first one on combining epidemiological and genomic data for inferring the source and spread of rabies is not supported. Unfortunately, the example of Rompon poorly supports that statement. Second, the implementation of IBCM together with genomic surveillance proved to be beneficial. The role of IBCM is simply not documented in that study. Third, the authors comment of the immediacy of the RTD results. However, the results show a delay of several weeks between bite and rabies confirmation. Forth, the lack of international guidance is not an obstacle. They should be included in the process of rabies diagnosis but confirmed by a recommended test. The WHO recommendations are clear. PEP should be started immediately.

Line 404: I agree that the lessons from the Romblon outbreak are that strengthening surveillance is key in monitoring a disease free area and in case of weak surveillance, rabies spreads rapidly. However, I do not quite see the benefits of One Health approach in the data presented.

Line 414-415: Suggest adding details of how IBCM was implemented - i.e., key steps with specific timeline, either directly in the methods or as supplementary information.

Line 421: Need to describe/define ABTC for the broader readership.

Line 423: Was this clarification based on previously published work? An animal killed by the community could be for various different reasons unrelated to rabies.

Line 425-426: Need to describe/define these different actors and at what spatial scale they intervene. Perhaps in the form of a diagram or glossary of terms.

Line 430: Why 10 days? Need reference.

Line 428-443: Need references for all the tests used.

Line 446: Why were only 24 sequences included?

Line 474-479: Need to provide more details, it's not clear how this simulation was done (e.g., how were outbreaks simulated? What assumptions were made?, etc.).

Line 498-499: Need to explain more the difference between the 32 scenarios. Other than the difference between the 16x16 scenarios, unclear where the other differences lie.

Reviewer #4

(Remarks to the Author)

Version 1:

Reviewer comments:

Reviewer #1

(Remarks to the Author)

I have no further comments and suggestions.

Reviewer #2

(Remarks to the Author)

Thank you to the authors for their extensive revision of the manuscript and responses to comments. In my opinion, the manuscript has been greatly improved by the additional context provided in the revised version. My comments have been appropriately addressed, however the final manuscript may benefit from a final revision for readability in places (e.g. Line 68 - 70 / Line 122). Changes that have been made to figures make the work easier to understand and additional information provided throughout the manuscript give greater clarity to the methods and results. I believe that the study now provides an important contribution to the literature in how the approach to enhanced surveillance in Romblon Province can provide guidance to developing similar methods for rabies control in other LMICs.

Reviewer #3

(Remarks to the Author)

Very well revised manuscript. However, there are still some points listed below that should be clarified or modified.

Page 3, general comment about the introduction: Given the broad readership of Nature Communications, I suggest having the first paragraph being more general about the application of genomics and epidemiology for zoonotic disease research and then zooming into rabies in the 2nd paragraph of the introduction.

Page 4, line 83.

In terms of phylogeny informing transmission dynamics, etc. , it would be worth citing there some other instrumental works and reviews published by other groups. I listed below some examples:

- Mathematical modelling and phylodynamics for the study of dog rabies dynamics and control: A scoping review. By Layan et al., 2021. PMID: 34043640
- On the Use of Phylogeographic Inference to Infer the Dispersal History of Rabies Virus: A Review Study. By Nahata et al, 2021, PMID: 34452492
- Using phylogeographic approaches to analyse the dispersal history, velocity and direction of viral lineages - Application to rabies virus spread in Iran. By Dellicour et al., 2019 PMID: 31535448

Page 6 line 137.

The authors refer to probable rabies cases in dogs “based on epidemiological evidence”. Could they provide details on how this probability was determined. Was it on clinical grounds? What is the rationale of this classification? Is this documented/validated in the literature? Need clarification on what “epidemiological evidence” refers too here.

Page 7, line 156-157.

If I am not mistaken, there are 31 samples tested by RDT and one of them is false negative. Therefore the sensitivity should be 96.8% instead of 95.7%.

Page 7, line 164. Suggest changing to: “...the vaccination year AND THE TYPE OF VACCINE WERE unspecified.” It is known that different vaccines may have different potencies. This should be relevant for rabies vaccines for veterinary use.

Page 7, line 194. The authors mentioned “Information, Education and Communication (IEC” activities. These are not described in the M&M. Please describe or give a reference.

Page 8, line 211: I suggest adding: However, “lack of OFFICIAL” recognition...”

Page 9: Fig2

Legend. Two human cases were not confirmed. Explain the basis on which these cases were considered as rabies cases. See similar comments on suspected animal cases above). Was this performed on clinical grounds, verbal autopsy...?

Fig2a. There are many human deaths reported. It would be nice to separate those that were laboratory confirmed from those that are only suspected cases.

Page 11. Lines 269 to 293. There is large uncertainty on the introduction time (Jul2020 to Jun 2022 for cluster 1 and from Sep 1985-Jul 1994 for cluster 2. Why is the uncertainty so important? At least the authors should mention and discuss this weakness. Further how was then the detection delay calculated considering these uncertainties? How do these uncertainties impact the simulation estimating the undetected cases? This calculation is not described in the M&M.

Page 13, Fig. 4: Panel B and C do not match with the legend. Need to switch them around. Additionally, many thanks for

adding the Supplementary Fig. S4, that is a great addition and makes the approach a lot clearer. However, a minor suggestion would be to increase the size of the panels and make the text less blurry in the figure. I found it hard to read.

Page 13, line 335. Given that there is no causal effect between rabies introductions and decreased vaccination coverage associated with the COVID-19 pandemic, I suggest the authors tone down this argument, particularly in the abstract, because it implies it is a main finding of the study when it is more of speculation. Additionally, here the authors indicate that long-distance human-mediated translocations could be a contributing factor, but do not mention it in the abstract. Along those lines, have the authors considered that an increase in rabies cases post-pandemic could be linked simply to greater surveillance with the introduction of the IBCM program during that time? Perhaps worth discussing in the Discussion. I also think there needs to be a discussion about the difference in vaccination coverage pre- vs post-pandemic (pre- ranging from 18-38.6% vs post- ranging from 0-24.2%) which arguably may not be different enough to influence rabies transmission in this system.

Page 14, line 343 and Page 16 line 454. It might be difficult for the reader to understand why 60% of vaccination rate is mentioned on page 14 and 70% on page 16. Please explain or homogenize.

Page 14, line 375. To describe the use of RTD, the authors refer to 3 publications. It would be worth to cite the original publication from 2016 who describe the use and the interest of this test in developing countries: Validation of a Rapid Rabies Diagnostic Tool for Field Surveillance in Developing Countries by Léchenne et al., 2016 PMID: 27706156.

Page 15, line 393. Mentioning other emerging diseases seems a little out of context here since most of the manuscript is focused on rabies only. Suggest saying more about other diseases earlier or cutting this.

Page 17, line 475. Border control measures are not addressed in the paper. Please give a reference dealing with this matter.

Page 18, Line 519. It would be nice to cite here (see my previous comment): Validation of a Rapid Rabies Diagnostic Tool for Field Surveillance in Developing Countries by Léchenne et al., 2016, PMID: 27706156.

Page 18, line 526. Saliva and nuchal skin biopsies are not commonly used for the laboratory diagnosis of infectious diseases. Please explain as this is not described in the M&M. Give a reference. I suggest some examples below:

- A reliable diagnosis of human rabies based on analysis of skin biopsy specimens by Dacheux et al., 2008, PMID: 18937576
- Diagnostic tests for human rabies by Dacheux et al., 2018 PMID: 30747123

Page 19, lines 539-542: Suggest adding a reference to back-up this approach.

Reviewer #4

(Remarks to the Author)

Reviewers' comments:

Reviewer #1:

1.1 The article entitled “combining genomics and epidemiology to investigate a zoonotic outbreak: rabies in Romblon Province, Philippines” provides a retrospective description of a rabies outbreak in a Philippine archipelago, that was considered dog rabies free for less than a lustrum, using classical descriptive epidemiology methods combined with phylogenetic analyses. Authors described the temporal spatial distribution of rabies cases during the recent outbreak 2022-2023 (using 43 positive animal cases and 2 humans), which were placed in context with historical information on rabies cases in animals and humans, animal vaccination coverage gathered since 2003 and 2011, respectively (figure 1). Authors used an existing integrated bite case management system couple to laboratory-based rabies diagnosis as a sentinel proxy for the early detection of potential rabies outbreaks in the Philippines. To have a more detailed insight on how rabies could have appeared in Romblon province, authors used almost complete rabies virus genomic sequences for 23 animals (with a genome coverage ranging from 90 to 99%) and 1 human (with 88% genomic coverage), which were put together with a contextual historical and contemporary data set of 746 unique sequences obtained from 1998 to 2023. In general, authors used sound and recent methodologies for both epidemiological and phylogenetic analyses, which allow them to infer the rabies outbreak studied was likely a consequence of at least three independent introductions from multiple ongoing rabies foci across the Philippines. Highlighting, how well interconnect is this topographically highly complex archipelago. The phylogenetic reconstructions analyses reveal several potential gaps in conventional surveillance systems. One of the most important ones is a detection delay of rabies cases in the main reservoir host (in this case dogs), or in vectors (cats) inherent to the detection threshold inflicted by a lack of a 100% surveillance coverage across the urbanization gradient (encompassing highly populated areas to marginalized or isolated regions), the overall public awareness on rabies, and limited resources allocated for laboratory-based disease detection, and use of less sensitive “more affordable” laboratory-based diagnostic methods. The discussion is compelling and complete, including all serious gaps within the overall investigation.

We thank the reviewer for commenting on positive aspects of our work.

1.2. The conclusions reflect most of the findings although some of them are ambiguously generic public health statements not necessarily based on the results of this investigation.

We appreciate the feedback and have revised the manuscript text to convey our conclusions more clearly in relation to our results. Specifically we report important challenges to sustaining rabies freedom in archipelagic, rural settings, that relate to surveillance and outbreak response in the context of One Health:

- 1. Routine enhanced surveillance needs to be in place for early outbreak detection, to support rapid response including sensitization of communities to reduce the risk of human deaths (Lines 360-370, 471-474, 481-483):*

- a. *Use of IBCM can improve case detection and outbreak investigations. In our study, multisectoral collaboration was crucial to the detection of most cases that were found by investigating suspicious animals first reported in bite incidents by health workers treating victims (Lines 80-84, 104-107, 121-123, 146-152, 204-206, 372-274, 388-390, 463-465).*
 - b. *Genomic sequencing can be used to identify the source of an outbreak which may affect the targeting of control measures (Lines 84-88, 387-393, 463-465, 476-479). Preventative mass dog vaccination should be targeted toward rabies-endemic areas, that act as recurrent sources of incursions (Lines 58-65, 387-390, 458-459, 474-479).*
 - c. *RDTs can potentially speed up outbreak detection and inform local responses, but lack of international guidance on the use of RDTs for rabies disincentivizes their use. In the context of the Philippines, we found this contributed to delays to outbreak declaration and community sensitization (Lines 77-80, 204-212, 374-378, 419-430).*
 - d. *Risks of inter-island spread outside of formal border control measures need consideration, e.g. via commercial ferries or private pumpboats (Lines 182-187, 353-355, 471-476).*
2. *Establishment of One Health outbreak response capacity is required at local community, regional and national levels.*
- a. *While LMICs like the Philippines should have capacity to respond to rabies outbreaks, the disease is not prioritised by veterinary services in favour of diseases considered to be economic impactful (e.g. African Swine Fever). This capacity will be essential in the context of other emerging disease priorities (Lines 92-94, 395-401, 471-474).*
 - b. *Outbreak response must be deployed after detection of animal rabies cases rather than only after human deaths (Lines 426-430, 469-471)*
 - c. *Responses should be sufficiently large and fast (coordinating at relevant spatial/ administrative scales) as outbreak spread beyond the initial area of case detection is inevitable, given delays in both detection and response (Lines 197-200, 398-407, 453-458, 474-476).*

1.3. Suggestions and recommendations.

Introduction section

Lines 57 and 90. In line 57 authors refer dogs as the primary vector for rabies, being responsible for the transmission of rabies to humans. However, in line 90 mentioned the use of genomic surveillance to decipher reservoir dynamics. In general, the term vector is used to refer an intermediary necessary to maintain the life cycle or transmission cycle of a parasite (from viruses, protozoa or helminths). However, the most common term used across the rabies field is reservoir host, since different rabies virus (RABV) clades or variants are maintained in a independent fashion by specific species of mesocarnivores worldwide or by different bats species (only in the Western Hemisphere). Dogs are the primary and most widespread reservoir host of several distinctive major and minor RABV clades, which depict circumscribed/unique

geographic distributions across the globe. Only in very rare instances, dogs and cats may work as vectors of RABV variants established in other major rabies reservoir hosts such as vampire bats. Thus, authors should harmonize their terminology referring to dogs as a major rabies reservoir host and not as a vector, since that is what dogs are in the Philippines major rabies reservoir hosts and not vectors.

We have removed the term 'vector' (Line 28).

1.4. Results section

Phylogenetic inference subsection.

For this study authors produced nearly complete genomes for 24 samples, encompassing 23 rabid animals and 1 human, using 746 unique sequences with different nucleotide lengths (from 211 to 11797 bp) as an epidemiological contemporary and retrospective (historical) context to decipher RABV dissemination dynamics across the Philippines. However, authors do not mention anything about these contextual sequences. It is not clear whether all these contextual sequences only came from the Philippines, or there may be other sequences from neighboring countries. Thus, I consider pertinent authors should include a detail table indicating GenBank accession number, year, geographic location across the Philippines, species of each of these sequences, designated lineage, sequence length and reference if previously published. I do recommend a table using the same color codes for provinces as indicated in figure 2.

The contextual data constituted RABV sequences with the Philippines as the country of origin. These sequences belong exclusively to the Asian SEA4 clade, a phylogenetic clade entirely restricted to the Philippines (just 11 publicly available Asian SEA4 clade sequences are from outside of the Philippines and these are all introduced cases). We have added detail to the methods to explain this (Lines 545-547).

The metadata and sequences, along with the code used to clean, standardise and deduplicate the sequences and generate the phylogenetic tree is available in our GitHub repository (Lines 548-551), and we point to this resource in the phylogenetics section. The metadata covers the variables suggested by the reviewer (GenBank accession number, year etc) at each stage of processing, including the final standardised, duplicated form. Since this is a very large table we have not added this as a supplementary table, but are happy to do so if required.

We also apologise for a discrepancy in numbers throughout the manuscript, which we have corrected. The total number of sequence records sourced was 694, which reduced to 518 sequences after merging sequences with the same sample ID (i.e. different GenBank records for gene sequences from the same sample). This conflicts with the number the reviewer noted (n=746). Apologies for the confusion caused.

1.5. Lineage definition

There is no clarity in the criteria used for the definition of lineages, as well as the extraction of subtrees (Figure 2) oversimplifies the original topological characteristics better shown in the original trees.

Figure 2: Suggest adding the full tree as a first panel to show the 3 clusters (as opposed to putting it in supplementary materials)

Apologies that this was not detailed in the methods. We now elaborate on this from Lines 536-540: “A Romblon-only phylogenetic tree was generated in IQtree and Romblon sequences were divided into several phylogenetic lineages, for the transmission tree inference (see next section). These lineages were defined through patristic distance clustering with the adegenet package, using a threshold of 0.0004, which was selected by comparing patristic distance clusters with phylogenetic trees and considering the RABV evolutionary rate ($\sim 2 \times 10^{-4}$ substitutions/site/year).”

The threshold represents the amount of evolutionary change expected in 2 years (given a typical rabies evolutionary rate of $\sim 2 \times 10^{-4}$ subs/site/year) and should capture whether all the recent cases are connected or not given the timeline and history of rabies on Romblon. This includes sequences from the previous outbreak in 2011, published by Tohma et al. (2015, 10.1016/j.meegid.2015.12.001), that are part of the contextual dataset of 581 sequences. Note that this threshold enables us to cluster sequences that have high genetic relatedness (relevant in the context of reconstructing transmission chains in a local outbreak, for example) therefore it does not have the same relevance applied across the entire contextual tree from the Philippines. It was not our intention to characterise the full contextual tree here, as lead author Bautista is working on that analysis for her larger PhD study for which she sequenced all these viruses.

We believe the subtrees allow the readers to see the detail and context of Romblon sequences more clearly, which is the main purpose of the paper. The subtrees were extracted to include sequences from each Romblon lineage and at least 5 closest relatives from the contextual data. The original topological characteristics can be examined in Supplementary Fig S2.

1.6. In the case of lineage definition, authors mentioned they considered patristic distances as their main criteria for lineage definition, however there is no clear distance threshold they showed that delineates the inter-lineage and intra-lineage variation ranges. Authors should show a heat patristic distance matrix showing the average inter-lineages patristic distances, so readers can have a sense of the extent of genetic variation across linages.

We address the definition of lineages in our previous response (Point 1.5). We have added a heatmap of patristic distances as Supplementary Fig S3.

1.7. In regards lineage definition most authors first define statistically supported clades and subclades (high bootstrap values or posterior probabilities) as putative lineages across the complete trees, to later check if there is a correlation among sequences pertaining to specific clades and the geographic location they were obtained from. Lineages are usually color coded from the region or country where most of the sequences pertaining to that clade come from. In this study lineages where color coded differently, given the impression there is no connection between lineages and the geographic location they came from. It would be valuable for potential readers to see if there is a correlation among between clades and the

geographic region across Philippines they came from. Ancestral state determinations would corroborate such visual inferences.

For example, line 224, Authors refer to a “star-like phylogeny” or tree topology, when what is mainly depicted across Fig 2 subtrees looks more like polytomies in subtree 2 A1, while subtrees 2B1 and 2C1 may show a more star like topology. I think a star-like topology would be best appreciated the original trees shown in the supplementary figures if they are color coded appropriately.

Regarding lineage definition, we appreciate the suggestion to correlate clades with geographic locations, a common practice in many phylogenetic studies. However, our focus was not to characterise the entire diversity of rabies viruses in the Philippines at that scale of resolution (i.e. the 0.0004 lineage threshold). We have intentionally refrained from identifying lineages at broader scales as this is part of another body of work, which constitutes the PhD research of co-lead Bautista. This forthcoming work will include more focused phylodynamic analyses and will be published separately. In this study, we provide information on the geographic associations of the closest relatives of the Romblon cases, as depicted in Figure 2 with colour-coded strips matching colours on maps.

Regarding the reference to the "star-like phylogeny," we appreciate your observation that subtree 2 A1 also exhibits polytomies. We have adjusted the text accordingly (Line 264-267), and hope this revision aligns with your interpretation: "This cluster shows several polytomies, each with 'star-like' bursts. This pattern is indicative of an introduction from a common source, succeeded by numerous local transmission chains. The star-like signatures signify rapid dissemination within a naive population, making it unlikely that these cases resulted from sustained cryptic circulation on Tablas from the previous outbreak."

Reviewer #2 (Remarks to the Author):

2.1. This paper reports the detection and response to a rabies outbreak on an island province in the Philippines, making use of modern whole genome sequencing technology to gain insights into rabies introduction events, viral transmission, and rates of case detection. The study demonstrates the fundamental importance of strengthening One Health surveillance processes to progress national rabies control initiatives, as well as the global agenda for the elimination of dog-mediated human rabies by 2030 (Zero by 30). The experience reported in the use of rapid field-side tests is of benefit to increasing the detection of rabies outbreaks and evaluating rabies burden at scale in resource limited settings.

We appreciate the author's well-articulated feedback on our work.

2.2. The study presents a case for the enhancement of rabies surveillance through the use of modern gene sequencing technology, however more context is required

around the feasibility of establishing this capacity as a routine component of rabies surveillance in most endemic settings.

In our findings, we report an array of challenges to enhancing surveillance for rabies, including genomic surveillance (for example, from identifying clinically rabid animals, collecting samples in a timely manner, transporting them between islands to a laboratory equipped with sequencing technology, Lines 360-366) as well as the ways these challenges were circumvented (e.g. through IBCM, Lines 366-370, 372-385) to illustrate that it was effective and possible to implement these approaches in the Philippines. We do, however, also acknowledge ways in which surveillance capacity, particularly the use of genomic technology must be further developed (Lines 390-393). We believe these capabilities and approaches are broadly applicable to other emerging diseases and that developing them in the context of rabies will not only greatly build One Health and outbreak response locally, but will have important tangible benefits for public health and animal health.

2.3. The paper emphasizes the benefit that real-time genomic surveillance brings to the effective design and implementation of rabies control interventions, and uses the comparison of Ebola, Lassa Fever and Mpox (Line 87). It would be pertinent to discuss the operational methods and challenges of implementing whole genome sequencing in the study site and establishing real-time genomic surveillance more broadly. For example, it would be helpful to report in the methods how often sequencing was performed and whether it has been possible to incorporate sequencing as an ongoing component of the rabies control program in the Philippines.

Thank you for this comment. As per our response to the point above, in a resource-limited setting like the Philippines, implementing genomic surveillance is a challenge as are many aspects of One Health. We have now added more details to the results and methods concerning the genomic aspects of the outbreak investigation, highlighting challenges and how they were addressed (Lines 232-243, 432-439, 532-534).

2.4. It would be beneficial to report the average time between laboratory rabies diagnosis and identification of virus lineage in this study, as this is of relevance for WGS to be used as a tool in guiding outbreak response. If routine in-country genomic sequencing is yet to be established, it would be important to discuss why this is the case and whether widespread use of routine genomic surveillance is likely to be feasible in most rabies endemic settings within the Zero by 30 timeframe. Practical guidance on when genetic sequencing is most useful and possible ways of maximizing its benefit in situations where only a fraction of samples can be sequenced would be helpful.

We anticipate that targeted deployment of genomic surveillance for investigating outbreaks can contribute to the 'Zero by 30' goal. Outbreaks in putatively rabies-free areas are likely to increase in frequency as countries aim to increasingly secure and maintain rabies-free areas.

In our study, we use whole genome sequencing for the purpose of investigating the probable origins of the outbreak, the timing of introduction(s)

and to inform stakeholders of recommended control strategies. Identification of a virus lineage or the lineage it is most related to among the circulating lineages within the area through whole genome sequencing after laboratory confirmation is technically possible within 3 days. In our study, sequencing is performed in batches to complete one run with the most economical amount of samples as mentioned in the methods section (Lines 532-534). We were able to sequence the first three samples from the outbreak cases 3 weeks after the first case was detected. Two more sequencing runs were done thereafter (Lines 238-243). We considered 24 samples as sufficient for initial genomic analysis to discern the origins of the virus introduction and how the cases are related to each other.

The ability to attribute the source of the outbreak requires sequencing to have previously been undertaken from samples from across the country. This may be a limitation for some endemic countries, but is certainly achievable for countries like the Philippines, where sequencing has become increasingly accessible post-pandemic, but where implementation challenges remain. In countries with less capacity than the Philippines and where rabies is far from controlled, we argue that now is a good time to characterise local circulation through bulk sequencing of archived samples or those accumulated with improved surveillance capacity, as countries work towards '0 by 30'. This is an important prerequisite to being able to identify the source of rabies outbreaks in future and an area where there is considerable scope for researchers from higher income countries, to support and train researchers in LMIC settings. This approach could have mutual benefit for all involved without taking resources away from urgent public health and disease control measures. Moreover if genomic surveillance is not deployed for diseases like rabies, the technology and interpretation will remain difficult and unfamiliar in the context of emerging diseases of more concern to high-income countries.

In our revision we advocate that countries take steps towards building genomic capacity. Despite progress during the pandemic, large discrepancies remain between genomic capacity in high versus low-income countries and inequities will widen excluding LMIC scientists from global health conversations unless these approaches are used for pathogens posing immediate threats, such as rabies. We are more explicit in acknowledging the constraints and challenges to genomic surveillance in the Philippines, which we expect to be similar in many rabies endemic countries. We also include practical guidance on how rabies-endemic countries like the Philippines, can mobilise limited resources to generate impactful results most cost-effectively (Lines 387-393, 463-467, 470-477). Finally, we challenge a more global readership in asking how they can be overcome since many of these obstacles require more global equity, for example to improve supply chains for sequencing reagents and consumables for LMICs.

2.5. The authors mention the currently inadequate capacity for field-investigation/ vaccination and laboratory diagnosis of all submitted cases. Therefore it would be important to frame whether it would be more pertinent to direct limited funds towards establishing widespread investigation and testing of suspect rabid animals, before prioritizing the development of viral sequencing, or whether there would be possible

scenarios where advancement of both could be feasible. These constraints are significant when advocating priorities for advancing towards Zero by 30 and in enabling stakeholders in rabies control elsewhere to consider how genomic sequencing may be of relevance in their setting.

We agree that there is an urgent need to prioritise establishment of widespread investigation and testing of suspect rabid animals while simultaneously planning for the gradual implementation of viral sequencing. Indeed, this was our ethos for writing this manuscript which is based upon an implementation study of IBCM.

Specifically, we now better argue how use of RDTs could empower local practitioners and incentivise field investigations, particularly in island contexts lacking laboratory capacity and with limited inter-island transport. Given the high specificity of RDTs for rabies, laboratory testing capacity could potentially be prioritised for ambiguous cases with clinical suspicion but negative RDT results. Similarly, targeted sequencing of priority cases could maximise the impact of genomic surveillance and make optimal use of available resources. This kind of balanced approach could address immediate needs whilst building genomics capacity that is transferable for other pathogens and leading to a comprehensive genomics-informed rabies surveillance system to support the achievement of the Zero by 30 goal. We elaborate on this in our response to point 2.4 above and in our discussion (Lines 387-393, 476-479).

We also note that laboratory capacity for testing and molecular characterization and field capacity for sample collection and outbreak investigation come under different government sectors and are not generally competing with each other for resource i.e. RADDLs come under the Department of Agriculture , but the field investigators (Municipal Agriculture officers) and provincial veterinary officers come under local government, whereas RITM comes under DoH. However, this highlights the complexity of resourcing and how easy it is for the responsibility of rabies to be passed to other sectors. From the perspective of rabies control, dog vaccines are allocated by BAI to the poorest municipalities (GIDAs, Geographically Isolated and Disadvantaged Areas), but higher income municipalities are supposed to source these vaccines themselves. In practice this limits rapid coordinated response. Hence, a key recommendation from this study is to develop policies for achieving this coordination across sectors and geographies as needed for rabies control (Lines 453-459, 474-476).

We also better support these statements in the Discussion, elaborating on points such as:

“ The concurrent emergence of ASF prompted government-mandated enhanced surveillance across several provinces, including testing, culling, and banning importation of pork products from affected islands. In comparison, rabies outbreak response was decentralised, fragmented, differing between municipalities, and limited in scale. Prioritisation of ASF by the animal health sector set back investigations, and case confirmation delays slowed the public health response. No formal declaration of an island-wide outbreak was made,

and while few municipalities declared a state of emergency, rabies control was limited.” (Lines 396-402)

2.6. Greater description of the relationship between insights from gene sequencing and vaccination activities would help to showcase the working impact of the tools described. Whilst the discussion mentions the need to improve the initiation of a timely, coordinated response following the detection of rabies cases, the significance of genomic sequencing on guiding field strategy and vaccination response in this outbreak was not clear.

We thank the reviewer for this helpful comment. We now explain the significance of genomic sequencing in the context of a rabies outbreak, in identifying sources of introduction and optimising vaccination response as stated in Lines 350-351, 387-390, 474-477).

2.7. It appears that municipal-level data on vaccination output was not available, but some information on the scale and distribution of vaccination effort is reported in the discussion (Line 327). Some information is given about vaccination data availability in the caption of Figure 1 (Line 197 – 198), however this seems misplaced and availability of vaccination data across the study period should be included in the methods. The results would be improved if it were possible to include data on where and when vaccinations took place on Tablas during the study period in relation to local rabies case incidence.

Self-reported municipal-level data is included as Supplementary Fig. S1. On the recommendation of the reviewer, we have expounded on the extent of vaccination data collected in the methods (Lines 491-494). Details on vaccinations initiated as part of outbreak response can be found in Lines 190-202, 343-345, 402-407, 414-416, 453-456.

2.8. Transmission of lineages 2, 3, and to some extent lineage 1, appear to have only sustained from Oct 2022 to Feb 2023, however there is no mention of why these groups then appeared to disappear by March 2023 (Figure 3). Did a vaccination campaign take place towards the end of 2022 in the regions of Tablas where these groups circulated, or was it more likely that these lineages became extinct in the absence of a recent campaign?

Sequencing has not been completed for samples collected after March 2023 (Lines 232-233, 238-343, 532-534). We will be able to confirm whether these lineages are continuing to circulate once we complete another sequencing run on the samples that have since accumulated. We now make this clear in our discussion (Lines 440-444). However, we would be very surprised if the lineages we reported have gone extinct given only limited dog vaccination carried out in 2023. Our process mapping work with stakeholders has supported efforts to ramp up dog vaccination in 2024 and we hope to detect the impacts of these efforts on both rabies cases and circulating lineages in the near future.

2.9. The study by Bourhy 2016 (DOI: 10.1371/journal.ppat.1005525) reporting natural extinction of rabies virus introductions in CAR and is probably more relevant than the Ebola reference used in Line 308. The need for improved quality and

access to data on dog vaccinations would seem to be an important learning from this outbreak to better coordinate the vaccination response.

We replaced the Ebola reference with a reference to Bourhy's study (Lines 352, 737-738). Thank you for this helpful suggestion.

2.10. The clear presentation of spatial information to support outbreak response and campaign planning is core to the premise of the paper, however variation in the formatting of cartographic figures is currently disorientating. Keeping styling and extent of the Tablas maps consistent between Figure 1(D), Figure 2 (A2/B2/C2) and Fig 3(C) would aid the reader in orienting the maps between figures.

The figures have now been adjusted to include consistent map orientation and colouring. We hope our revised figures do a better job of communicating findings in a clearer way.

2.11. Figure 1C provides useful context to the study site (Tablas island) in relation to surrounding geography, but this comes after surveillance data are being presented in parts A, and B, so the reader has to jump around a bit to work out where cases are occurring. An initial figure providing the key geographic locations and boundaries would be a useful reference for understanding the spatial context of data presented throughout the paper. Perhaps Figure 1C could be pulled out as its own figure with labels for Romblon province and the text labels for RADDL, RITM and Manila made clearer.

We address this comment with the newly revised figure.

2.12. Santa Maria is referred to repeatedly in the paper, but this is not explicitly labelled anywhere in the figures, so perhaps a reference map of Tablas island as part of this figure also showing the key ports and other locations referenced in the results (Odiongan, San Agustin etc) would aid interpretation of the surveillance data.

We have revised the maps so that Fig. 2 has the municipalities outlined and Fig. 1 colours each municipality within Tablas island.

2.13. Specific comments:

Line 61 – The figure of 161 daily global rabies deaths appears to be calculated from an estimate from almost ten years ago and there have since been lower estimates (e.g. Gan et al 2022, 10.1016/j.ijid.2022.10.046). Please update the statement to more accurately reflect the reference and that this was an estimate.

We have now deleted this calculation and focus on more substantiated data from the Philippines, given contention around GBD estimates that lack data on rabies deaths from endemic countries.

2.14. Line 75 – The reference used to support the statement about incursions / importations across borders occurring regularly worldwide is a study focused on a single city (Arequipa). Please add references to support the worldwide nature of the comment.

Additional sources have been added to show that rabid animal importations and cross-border incursions occur regularly worldwide (Lines 67, 655-667).

2.15. Line 96 – Please check the sentence for correctness “...detection of a rabid dog last 2022 on the island of Tablas...”. Is the part “...last 2022...” a typographical error – should it read “...in 2022...”?

The sentence has been amended and now states “...detection of a rabid dog in 2022...” (Line 101)

2.16. Line 290 – Is there a source reference for the annual rabies deaths in the Philippines?

We are grateful to the reviewer for pointing out this oversight and have added a reference (Lines 338, 726-727).

2.17. Line 293 – It would be relevant to not only state the recommended target annual vaccination coverage to control rabies, but also to emphasize the importance of homogenous coverage across contiguous populations to realize elimination. It is mentioned earlier in the paper that vaccination activity was not equal across all municipalities, which could significantly undermine the likelihood of achieving elimination (Ferguson 2015 (DOI: 10.1038/srep18232), Townsend 2013 (DOI: 10.1371/journal.pntd.0002372), Suseno 2019 (DOI: 10.20506/rst.38.1.2954)).

We are thankful for your recommended reference (Ferguson 2015) and have included it as a source (Lines 343, 734-735). We have highlighted the impacts of heterogeneous vaccination coverage in Lines 341-343, demonstrating how this applies to Romblon in Lines 453-456 because of the lack of coordination between municipalities on conducting dog vaccination.

2.18. Line 347 – It seems of limited benefit to advocate for enforcement of dog leashing in a locality which has neither sufficient infrastructure nor enforcement capacity, and also where this practice is counter to cultural norms. Perhaps include a greater emphasis on the need for owners to take responsibility for the public health impact of dogs in their care, by ensuring that dogs are vaccinated and do not cause a public nuisance.

We appreciate the reviewer’s suggestion and have adjusted the text to include the following: “Therefore, the burden must also be shared with dog owners to take responsibility for ensuring that their pets are vaccinated and not inconveniencing others.” (Lines 450-451), and have removed the phrase referring to ordinances reducing bite incidents if implemented.

2.19. Line 375 – Is use of the word ‘parsimonious’ appropriate in this context? The definition of parsimonious describes someone unwilling to spend money, which is confusing when applied to this sentence.

We have removed the word ‘parsimonious’ as a descriptor (Lines 440-444).

2.20. Figure comments:

Figure 2 – This figure shows a lot of useful information, however is currently

challenging to understand across all the components. The positioning of the fill color legend for lineage above the phylogenetic trees (A1 – C1) and for province above the map (D), make it initially difficult to understand that the first color strip in the tree figures relates to the Province fill legend above the map on the other side of the figure when reading from left-to-right. It might be clearer to either put the province fill legend first in the top left, followed by the lineages legend, but perhaps with the legend title as “Romblon lineages” to make it clear that these are the cases from Romblon. It would be helpful to add a reference point to the province map to help orient where Tablas is. The Tablas maps could be cropped to match the extent in Figure 1(D), which would enlarge the island to show the locations more clearly. The color of Romblon province in parts A2-C2 appears to be a lighter shade of that in the province map, however this introduces another color that does not match any of the other labelled colors. You could change Romblon in the province map to the lighter color for consistency throughout the figure.

Thank you for your suggestions to improve Fig. 3 (formerly named Fig. 2) aesthetics, we have addressed as follows:

- *The order of the colour strips in panels A1-C1 has now been switched showing the bar for Province first, then lineage. This corresponds to the order of the legend at the top of the figure.*
- *We have changed legend title to “Romblon lineages”*
- *The bounding box for panel D now matches Figure 1C, similarly the cropped extents of Tablas maps in panels A2-C2 now match Figure 1D.*
- *Have added text to orient to Tablas island*
- *The colour in A2-C2 is the same as the province map except that transparency has been added to allow points to be seen clearly. We have now added the same transparency across the other panels so the colour shading remains consistent.*

2.21. Figure 3 – I’d suggest putting the legend fill color label in the top left corner of the figure so that it is the first thing seen when reading for the first time left-to-right. Sub-labels of A / B / C are not currently shown within the figure, but are referenced in the caption. It is slightly confusing that colors are conserved between Figure 2 and 3, but icon shape convention is not consistent – in Figure 2 circles are sequenced cases and squares are inferred province-level ACR location, whereas in Figure 3 squares are sequenced cases and circles are inferred lineage of un-sequenced cases. Could the squares / circles be reversed in one of the figures to help with continuity in designation of sequenced cases across the figures?

We have incorporated the reviewers suggestions in the newly-revised figures.

2.22. Reviewer #2 (Remarks on code availability):

The datasets all appear present and the code appears sufficient to reproduce the results and figures presented in the manuscript. There is a README file with sufficient instruction and additional guidance in the R/overview_tmtrees_scripts.docx file. Whilst I have not reviewed all of the script files, the few that I have run are working well with the data provided.

We are appreciative of the time and effort spent in testing that our script files run smoothly.

Reviewer #3 (Remarks to the Author):

3.1. General comments

Yuson et al. present an interesting paper describing the problems encountered to investigate and mitigate a zoonotic outbreak of rabies in Romblon Province, Philippines. They suggest that combining rapid whole genome sequencing (WGS) and epidemiology of rabies viruses (RABV), one of the major neglected tropical disease causing a large public health and economic burden in Africa could reduce the lag time from sample to epidemiological action in LMICs and improve the implementation of control measures by providing a better and more rapid understanding of disease transmission dynamics.

The article is well written, the findings are important for guiding rabies surveillance and control measures in the Philippines, and the study is a nice illustration of the importance of combining IBCM and genomic surveillance for rabies outbreak investigations and further demonstrate the importance of prompt implementation of control measures to limit the spread of rabies after re-introduction in a rabies-free area.

We appreciate the detailed, positive feedback from the reviewer.

3.2. However, the article is written in a way that gives the impression that the approach is novel when in reality several other rabies studies have used similar approaches, which were not referenced in this manuscript.

The concept of this paper has already been presented in a previous paper co-signed by some of the same authors and showing how the WGS and the management of genomic data could play a role in a surveillance context in a different country (see Lushasi et al., *Elife*. 2023 May 25;12:e85262) .

Several of the manuscript co-authors have used WGS methods to investigate a rabies outbreak on Pemba island off the coast of Tanzania. However, we hope that does not negate the value of our work in the Philippines. Here we draw from a recently published method for combining partial and whole genome data (Holtz et al. 2023) for the contextual tree, we use a new approach advocated for One Health surveillance of rabies (Integrated Bite Case Management), supported by use of RDTs, as well as the WGS and transmission tree methods used previously. Moreover, the situation and context of the outbreak is novel (a rural, archipelagic setting within the Philippines where rabies is generally under better control than in settings in sub-Saharan Africa), as are the insights gained, and the interpretation. In our revision of the manuscript we have brought out these important differences.

3.3. In this study, the authors claimed that they demonstrate the interest of real-time sequencing of RABV to rapidly inform policy decisions and disease management. However, this study neither demonstrates the feasibility of real-time sequencing of RABV in the setting of Romblon province nor does it demonstrate its impact to inform policy decision and disease management. This demonstration is impaired by many limitations (important delays between bite and laboratory diagnosis, difficulties in shipping the samples, lack of confidence in the processes, set up of new procedures after the re-introduction) that are presented in the discussion. These limitations dramatically weaken the ability of this approach to accurately inform epidemiological surveillance and complex epidemiological questions in real time and this study does not bring any significant clues to solve them.

We are sorry that the way we presented our work led to the reviewer's conclusions about real-time sequencing not being feasible or impacting policy or disease management. It was not our intention to focus on sequencing in real-time. We have therefore removed the reference to 'real-time genomic sequencing' and have clarified that it was tracking the investigation, from the first rabies-positive case, that was conducted in real-time. We have now expanded on many aspects of the enhanced surveillance that were undertaken and we clarify the limitations and challenges we encountered.

Specifically we expand on the RDT use for detection of the first discovered case and the ripple effects on finding and testing subsequent cases in Lines 126-135:

"The first detected rabies-positive case (November 21st, 2022) was a dog that was investigated three days after its involvement in a biting incident (November 18th) in Santa Maria municipality. This was the first sample from the province to have been tested for rabies since 2020, and the first local use of an RDT after being supplied for IBCM (training carried out in March 2020 just before COVID-19 restrictions were announced). Due to the absence of laboratory facilities in Romblon province and the fluorescent microscope being broken at the Regional Animal Disease Diagnostic Laboratory (RADDL 4B, Fig. 1b), the sample was transported overnight to the National Reference Laboratory at the Research Institute for Tropical Medicine (RITM) in Manila. Here it was confirmed the next day (November 22nd) through direct fluorescent antibody testing (DFAT) and the positive result was immediately communicated to the local government, prompting increased sample collection and in-field testing."

We also clarify how the enhanced surveillance informed policy decisions (Lines 195-197):

"That same month, the governor of Romblon Province instructed all municipalities' mayors on Tablas Island to enforce Republic Act No. 9482 (Anti-Rabies Act of 2007), requiring local government units to allocate funds toward dog vaccination."

To support the argument that our approach was used for, and can be used to inform disease management, we added the following:

“Sequencing has played a crucial role in informing sources of rabies introductions and mobilising vaccination responses.. Integrating genomic data with epidemiological data from IBCM enhanced understanding of the outbreak spread and identified possible points of introduction, also suggesting the need for preventive vaccination, targeting dog vaccination towards source endemic areas. The benefits of genomic surveillance, as evidenced during the COVID-19 pandemic, require that expertise and skills are maintained in-country. Applying these methods for rabies can help build and sustain capacity for outbreaks of other emerging diseases.” (Lines 387-393)

“Samples from animals involved in high-risk bites were mostly not collected prior to the first RDT-positive case, but the result sparked multiple investigations, leading to increased sample collection and RDT use. News of the case result also catalysed testing of two samples that had been stored for over a month. These laboratory-confirmed cases proved that rabies had been circulating earlier than initially presumed.” (Lines 366-370)

3.4. Further, some of the methodologies need to be described in more detail (such as the IBCM approach and the simulation work).

We appreciate this suggestion and have added Fig. 5 in the Methods detailing the IBCM process, to better explain the instrumental role it played in outbreak detection and case finding (Lines 507-513).

3.5. Finally, most of the conclusions are not supported by the data presented (see below comments on Line 393-397).

Line 393-397. The authors stated that their study demonstrates 4 different points. The first one on combining epidemiological and genomic data for inferring the source and spread of rabies is not supported. Unfortunately, the example of Romblon poorly supports that statement. Second, the implementation of IBCM together with genomic surveillance proved to be beneficial. The role of IBCM is simply not documented in that study. Third, the authors comment of the immediacy of the RTD results. However, the results show a delay of several weeks between bite and rabies confirmation. Forth, the lack of international guidance is not an obstacle. They should be included in the process of rabies diagnosis but confirmed by a recommended test. The WHO recommendations are clear. PEP should be started immediately.

- 1. In response to the reviewers first point “Combining epidemiological and genomic data for inferring the source and spread of rabies is not supported. Unfortunately, the example of Romblon poorly supports that statement” , we highlight in the text the benefits gained from combining epidemiological and genomic data.*

Thorough case finding efforts yielded samples in all but one municipality, showing the extent of rabies spread (which at that point, covered the whole island) by the time of outbreak detection. Genomic data showed that several incursions led to the outbreak, that were traced to other rabies-endemic provinces indicating that border control measures had failed to stop human-mediated movement of dogs via sea travel as discussed in Lines 350-358, 387-393, 432-444, 463-465.

2. *In response to point 2 “The role of IBCM is simply not documented in that study” we have added Fig. 5 to explain the IBCM process and we have amended the text to emphasise the instrumental role IBCM played in detecting the outbreak (Lines 126-151, 366-370) and subsequent case finding. Most identified cases were the result of health workers coordinating with their animal counterparts to investigate suspect animals reported by bite victims who were seeking PEP (Lines 146-152, 204-206, 372-278).*
3. *In response to the reviewers 3rd point “The results show a delay of several weeks between bite and rabies confirmation”, we have clarified in-text that the first detected case was confirmed rabies-positive within a week of the bite incident (see Lines 126-145, also detailed in response to point 3.3 above). We also clear up confusion about the testing delays for the two previously stored samples (collected in September and October, respectively). These were not tested until results of the first rabies-positive case was released in November: “News of the case result also catalysed testing of two samples that had been stored for over a month. These laboratory-confirmed cases proved that rabies had been circulating earlier than initially presumed.” (Lines 368-370, 426-428).*
4. *In response to “Fourth, the lack of international guidance is not an obstacle. They should be included in the process of rabies diagnosis but confirmed by a recommended test. The WHO recommendations are clear. PEP should be started immediately” we clarify that PEP was started immediately for all biting victims that reported to ABTCs regardless of whether an RDT was performed. Moreover, all the samples initially tested by RDT were forwarded for confirmatory DFAT testing, regardless of RDT results. It should be noted, however, the immediacy of RDT results helped expedite tracing of other exposed victims, as results of rabies-positive cases were disseminated immediately and therefore used to encourage other bite victims to come forward and seek PEP.*

3.6. In general, this is an important article but does not meet the novelty standards of a large audience journal.

Given that our study takes place in a novel setting using a range of new approaches and highlighting important challenges for One Health, we have rewritten the manuscript to better highlight these points (see response to Reviewer 1, point 1.2). We particularly focus on IBCM which we believe we didn't highlight sufficiently previously and we hope this is now much clearer.

3.7. Specific comments:

Line 29-30: suggest mentioning other aspects of the global strategy to eliminate human deaths such as improving surveillance and community participation.

We appreciate this suggestion; the revised sentence now states: “Zero by 30’, the global strategy to end dog-mediated human rabies, promotes a One Health approach underpinned by mass dog vaccination, post-exposure vaccination of bite victims, robust surveillance and community engagement.” (Lines 28-30).

3.8. Line 35-36: why put so much emphasis at this stage on the role of suspension of routine dog vaccination due to COVID-19 restrictions. This is not demonstrated in the paper.

We have highlighted examples of rabies re-emergence worldwide in Lines 50-52 due to neglect of dog vaccination. We also revised Lines 180-182 in Results to mention that the sharp decline in vaccination coverage was due to social distancing restrictions that limited mass gatherings and therefore prevented vaccination campaigns from proceeding.

Fig 2A also emphasises the COVID-19 lockdown period, showing that no mass dog vaccination took place during those years when previously vaccinations had been carried out each year. Lines 345-348, 352-355 and 363-366 in the Discussion mentions the COVID-19 lockdowns in the context of re-emergence of rabies on other formerly rabies-free islands in the wake of the pandemic.

3.9. Line 45-55: Suggest discussing rabies in the context of neglected tropical diseases at the start of the introduction rather than emerging infectious diseases. It could be put in the context of EIDs later on in the introduction when discussing re-introductions.

Thank you for this suggestion. We have revised this part of the introduction as recommended (Lines 45-52).

3.10. Line 47-49: There are many examples of re-introduction of rabies in areas where it was previously eliminated. They should be cited.

Several examples of rabies re-introductions are mentioned in Lines 65-69:

“However, introductions and re-emergence of rabies through animal importations by humans or from natural incursions across borders occur regularly worldwide. Examples from the city of Arequipa in Peru, Sarawak in Malaysia, and Mpumalanga province, South Africa, demonstrate how neglecting surveillance and dog vaccination can lead to rapid escalation from introductions in areas close to rabies-endemic zones.”

3.11. Line 50-53: Rabies also requires surveillance and community awareness campaigns, not just PPE and mass dog vaccination. Suggest also adding references here.

The sentence now includes mention of both awareness campaigns and surveillance, with a reference added:

“Rabies is fatal once symptoms appear, but progression to disease can be prevented if immediate post-exposure prophylaxis (PEP) is given to bite victims after exposure. PEP, while highly effective, should be part of a broader rabies management strategy which includes educational campaigns to increase awareness, robust surveillance for case detection, and mass dog vaccination to interrupt dog-to-dog transmission, thereby reducing the risk of human exposures.” (Lines 57-61)

3.12. Line 53-55: Suggest being more specific. The current arguments are quite broad making it difficult to really see what the barriers are.

The lines have been revised to be more specific:

“However, achieving successful rabies control requires overcoming challenges such as limited human resources and cross-sectoral financing. Government priorities typically favour investment in animal diseases that have economic impacts such as African Swine Fever (ASF), whilst political and economic instability with frequent changes in governance make programmes difficult to maintain.” (Lines 91-94)

3.13. Line 57-58: Primarily in Africa and Asia

We have amended the sentence to specify that human rabies deaths mainly occur in Africa and Asia (Line 50).

3.14. Line 61-62: Here it is worldwide but in the abstract it's only in LMICs.

Both the text and abstract now state that deaths mainly occur in LMICs in Africa and Asia (see Point 3.13 above).

3.15. Line 66-67: Could the authors provide any evidence of the role of dog vaccination in the elimination of dog rabies in Europe in the beginning of the 20th century. There may be a confusion here with the elimination of fox rabies at the end of the 20th century.

We have included a source that mentions the first use of veterinary vaccines in Europe/US post-WWII in the 1920s that severely reduced domestic animal cases in Europe (Lines 645-647).

3.16. Line 69-72: It's difficult to follow the reasoning for the examples selected. Suggest citing other well-known studies in African settings, in Chad for example.

We have added N'Djaména, Chad (Line 65) alongside Pontianak, Indonesia as an example of a local rabies-free zone. For the examples of recent re-emergences of rabies, we selected examples in Peru, Malaysia and South Africa as they reflect specific instances of delayed case detection and neglected dog mass vaccination in formerly rabies-controlled zones that, following an incursion from a nearby rabies-endemic area, led to a re-emergence of rabies. In particular, Indonesia and Malaysia were included because of their similar settings to the Philippines, due to their archipelagic nature and proximal location in Southeast Asia.

3.17. Line 73-79: It seems that there is a confusion here between mass vaccination of dogs to eliminate rabies from an enzootic area and to prevent re-introduction into a free area. Could the authors provide examples on how these strategies may differ in terms of % of dog vaccination coverage in various settings?

Our intention is to emphasise: (a) that lack of sustained rabies control measures, namely mass vaccination, in rabies-free areas, allows incursions to escalate into outbreaks and (b) if outbreak response is not swift nor

sufficiently expansive, outbreaks remain uncontrolled. We believe that the Romblon outbreak illustrates both points as vaccination coverage had sharply declined prior to the outbreak (Lines 343-347, 363-364) and ring vaccination as a response was insufficient for ending the outbreak (Lines 402-406, 474-476). We hope our revisions here are clearer now.

3.18. Line 82-87: Many tools and approaches are mentioned without giving any reference to rabies although there are many regarding the use of RDTs, mobile communication and sequencing. Instead there are many citations regarding other zoonotic diseases.

We have added more citations on other approaches used in rabies studies, like IBCM and genomic sequencing. We have also added more examples of rabies genomic surveillance (Lines 78-79, 82-88).

3.19. Line 85: Suggest defining “real-time” since the methodologies indicate multiple day delays until diagnostic testing and sequencing, suggesting that the approach is not real-time per say.

We are thankful to the reviewer for this suggestion and have removed the single reference to ‘real-time genomic sequencing’ (Lines 99-100). Our approach to the outbreak was a real-time tracking and investigation beginning with its initial detection in November 2022. We have subsequently gathered data for every bite incident, animal investigation, sample collection, RDT and confirmatory laboratory test as they happened, and continue to do so in 2024.

3.20. Line 85-91: Unclear why the others present genomic surveillance using other zoonotic diseases as examples when there is quite a lot of literature on rabies genomic surveillance. Suggest prioritizing the rabies literature.

Similar to our response to Point 3.18 above, we have focused on citations related to IBCM and genomic surveillance (Lines 82-88).

3.21. Line 130: At this stage, provide references on the use of RDTs for rabies diagnosis rather than the reference of the kit which can be given in M&M.

The previous reference has been replaced with journal articles that evaluate RDTs as a diagnostic tool for rabies (Lines 77, 678-685).

3.22. Line 137-139: These delays are long. Could the authors provide the delay between death of the animal and diagnosis? It should be around 14-15 days if I am not mistaken. What could explain such a long time? How could this rapidly inform therapeutic decision? What was the delay between exposures and PEP in the treated patients? How could the whole surveillance system be reactive and sequencing data be informative if such delays remain?

We agree that the average infectious period for biting animals is around 2 weeks (7-14 days). However, we observed two cases wherein infectious period exceeded that duration.

Delays between biting incident and RDT use were due to some cases wherein there were time gaps between sample collection and testing. We have

included more lines in the discussion elaborating on this (Lines 138-140, 366-370, 376-385, 419-426)

3.23. Line 141-145: These data are not very specific of this study and not very informative regarding the objective of the study. I suggest to present them in a supplementary table.

This suggestion has been duly noted and the sentences removed from the main text. We have decided to replace them with more relevant data (Lines 161-166, 174-175)

3.24. Line 145: How does % of rabies positive dogs reported as previously vaccinated compare to % of rabies-negative dogs? Is it significantly different? How does this compare with the estimation of % of coverage given for the province?

We have included further information on the rabies vaccination status of tested dogs in Lines 161-166.

3.25. Line 170: Does the confirmation occur in November regarding the delay of diagnosis (date of case + 23 days)?

The delays in diagnosis were in relation to the 2 samples stored in a freezer in Romblon since September and October, respectively, and only tested following confirmation of the November sample. There was minimal delay in confirming the November sample. We have attempted to clarify these details (Lines 138-140, 366-370).

3.26. Line 199 (Fig 1): What do the dates provided correspond to: date of death, date of diagnosis?

For humans, date of death was used. For animals, the date of the biting incident was used where known. Otherwise, the date of sample collection was used instead (for 22/43 cases). We now clarify this in the figure legend.

3.27. Line 207: 23 DFAT positive samples are mentioned but 43 were tested positive by RTD. Explain the difference?

All 43 samples tested using RDT, whether they yielded positive or negative results, underwent subsequent laboratory confirmation by DFAT. However, not all laboratory confirmed samples were sequenced; only the first 23 samples collected from October to March were sequenced (Lines 238-243). We have revised the text, which now states: "DFAT confirmed animal brain samples (23/25) had genome coverage of 90-99% while a lower coverage of 88% was achieved for the human skin biopsy sample that was confirmed by nested PCR." (Lines 234-236).

3.28. Line 211-212: Need to add information on sample sizes by year and geographical location. Same comment for sequence length and regions of the genome that were sequenced.

The metadata and sequences, along with the code used to clean, standardise and deduplicate the sequences and generate the phylogenetic tree is

available in the GitHub repository (Lines 550-551), which we also point to in the phylogenetics section. This includes the metadata suggested by the reviewer (GenBank accession number, year etc) at each stage of processing, including the final standardised, duplicated form. Since this is a very large table, we have not added this as a supplementary table, but are happy to do so if required.

3.29. Figure 2: Suggest adding the full tree as a first panel to show the 3 clusters (as opposed to putting it in supplementary materials)

Thank you for this suggestion. We are maintaining the full tree as a supplementary file for now because it is very large due to the considerable amount of contextual sequences used. We could collapse some of the tree branches to fit in a small panel in Fig. 3 (formerly named Fig. 2) and would do so if still required.

3.30. Line 223-237: It appears that there is lot of uncertainty regarding the number of importations given by the phylogenetic analysis and considering the fact that only 24 samples out of 43 positive ones were sequenced. This is expected and it is known that such a study could definitely underestimate this figure. The authors should then be more cautious. There are probably many more introductions than those reported by the phylogenetic analysis. This is one of the major limitations of the phylogenetic approach. For an illustration, see a study performed in Bangui in Central African Republic which clearly shows how underreporting, limitations in sample collection and the rate of evolution of the RABV genome impair the resolution of introduction events and the resolution of transmission chains based on phylogenetic studies compared to what can be evaluated by modeling (Bourhy et al., PLoS Pathog. 2016 Apr 8;12(4):e1005525)). Therefore, the detection delays calculated line 241 should also be given with caution.

Thank you for this comment. We appreciate the uncertainty in this kind of inference and have referred more to this in the discussion where we report limitations of the study (Lines 432-444).

3.31. Line 290: Need a reference here.

We are grateful to the reviewer for pointing out this oversight and have added a reference in Line 338.

3.32. Line 292: May the authors comment the present epidemiological situation of Bali regarding rabies despite mass dog vaccination. How does this compare with Bohol?

We have revised Lines 338-343 to include the statement: "Mass dog vaccination is effective for rabies control, and has been employed nationwide at varying consistencies. One successful local example is Bohol Province's intersectoral elimination program, which achieved 70% coverage (as recommended by WHO) through "catch-up" vaccination following mass campaigns. Models predict that vaccinating at least 60% of dogs should substantially reduce cases, but if coverage is heterogeneous, time to elimination increases, while the probability of elimination decreases." We

discuss Bali and Flores' epidemiological situations in the Introduction instead (Lines 69-72).

3.33. Line 305 "Our work and that of others": from a general point of view there are many interesting and important studies from other groups that are not mentioned. Suggest that the authors could make some efforts to mention other instrumental studies and reviews in that field.

We appreciate the reviewer's suggestion. We have now added references to other studies about genomic sequencing, IBCM, and One Health in the management and elimination of rabies Lines 77-88, 350-351).

3.34. Line 308: Again, there are studies dealing with RABV illustrating this concept. No need to refer to EBOLA (REF #32).

We have replaced the Ebola reference with the Bourhy et al. study (Line 352, 737-738).

3.35. Line 314-319: Data on submissions between 2011 and 2022 are not available. Please document. Illustrate the fact that the case detection rate increased from 2022.

We have included information on cases reported before 2022 in Fig 2a and now state, Lines 121-122, that the number of sample submissions ranged from 6 to 39 annually, dropping to <5 during the pandemic lockdown.

3.36. Line 386: Could the authors provide data showing "increased case detection due to IBCM promoted communication between animal and human sectors across several provinces". This statement is so far not supported.

We have revised the manuscript to better explain how initial detection of rabies (which was the result of IBCM-supported reporting of a suspicious dog from a bite victim seeking PEP) triggered widespread testing (Lines 126-140, 146-152, 366-370, 372-382). See also our response above to point 3.35.

3.37. Line 404: I agree that the lessons from the Romblon outbreak are that strengthening surveillance is key in monitoring a disease free area and in case of weak surveillance, rabies spreads rapidly. However, I do not quite see the benefits of One Health approach in the data presented.

To emphasise the instrumental role IBCM (the One Health approach we are referring to) played in tracking the outbreak in real-time, we have included more details on the initial case detection – which was triggered by a biting incident, leading to communication by the human health sector with animal health sector counterparts to mount an investigation – and how the positive RDT result stemming from this investigation spurred further animal investigations and sample collection (see response to Point 3.36 above).

3.38. Line 414-415: Suggest adding details of how IBCM was implemented - i.e., key steps with specific timeline, either directly in the methods or as supplementary information.

We have added details of how IBCM is implemented in the supplementary information, including a diagram of the process (Fig. 5).

3.39. Line 421: Need to describe/define ABTC for the broader readership.

The sentence has been revised and now defines ABTCs as “clinics in hospitals or health units that provide PEP to bite victims” (Lines 500-501)

3.40. Line 423: Was this clarification based on previously published work? An animal killed by the community could be for various different reasons unrelated to rabies.

The distinction between ‘dead’ and ‘killed’ animals is not based on previously published work. Probable rabies cases are classified as such if an animal is reported as dead under any circumstances. However, after initial IBCM training, some local health workers only investigated animals that had died of natural causes, and had to be informed that killed animals (by community members, vehicles, or other animals) were also classified as high-risk even if the cause of death was not explicitly due to illness.

3.41. Line 425-426: Need to describe/define these different actors and at what spatial scale they intervene. Perhaps in the form of a diagram or glossary of terms.

We now explain IBCM in detail in the methods (Fig. 5).

3.42. Line 430: Why 10 days? Need reference.

A reference has been added (Line 517).

3.43. Line 428-443: Need references for all the tests used.

References for RDTs have been added to Lines 77 and 518. Per the reviewer's suggestion, we have added a reference for DFAT also (Line 521).

3.44. Line 446: Why were only 24 sequences included?

The first 24 sequences were generated after 3 consecutive sequencing runs following outbreak detection, and covered cases detected in the first few months (from September 2022 to March 2023). This sequencing run was deemed sufficient for the purpose of finding out where the introductions originated from, or whether it was from previously circulating undetected cases related to the first outbreak from a decade ago. We have indicated our reasoning in the Methods. More samples have since been collected and another sequencing run is underway that prioritises new outbreaks on previously rabies free islands recently detected in the region. The additional Romblon samples are included in this run for more cost-effective bulk sequencing.

3.45. Line 474-479: Need to provide more details, it's not clear how this simulation was done (e.g., how were outbreaks simulated? What assumptions were made?, etc.).

Additional details are now provided on the branching process model and the assumptions made (Lines 284-293, 573-581).

3.46. Line 498-499: Need to explain more the difference between the 32 scenarios. Other than the difference between the 16x16 scenarios, unclear where the other differences lie.

Apologies for the lack of clarity regarding the scenarios. We are now more explicit in the manuscript text (Lines 595-609): “We generated 1000 bootstrapped trees for each of 32 scenarios, corresponding to all combinations of: (i) case locations (barangay centroids versus locations sampled from the population density grid); (ii) use of genetic data for inference (yes/no); and (iii) inclusion of pruning steps to further resolve transmission chains (eight different combinations of cut-offs). [...] In the scenarios with additional pruning steps, case pairs were filtered out where the time interval and/or distance exceeded specified percentiles of the serial interval and distance kernel distributions. Pruning options included no pruning, pruning by time only (cut-offs 0.95, 0.975 and 0.99), pruning by time and distance using the same cut-offs (0.95, 0.975 and 0.99) and one combination of differing cut-offs (time 0.975, distance 0.99).”

Reviewer #4 (Remarks to the Author):

4.1 I co-reviewed this manuscript with one of the reviewers who provided the listed reports. This is part of the Nature Communications initiative to facilitate training in peer review and to provide appropriate recognition for Early Career Researchers who co-review manuscripts.

This comment has been noted.

REVIEWER COMMENTS

Reviewer #1 (Remarks to the Author):

1.1. I have no further comments and suggestions.

Thank you, we are glad that we have addressed all concerns.

Reviewer #2 (Remarks to the Author):

2.1. Thank you to the authors for their extensive revision of the manuscript and responses to comments. In my opinion, the manuscript has been greatly improved by the additional context provided in the revised version. My comments have been appropriately addressed, however the final manuscript may benefit from a final revision for readability in places (e.g. Line 68 - 70 / Line 122). Changes that have been made to figures make the work easier to understand and additional information provided throughout the manuscript give greater clarity to the methods and results. I believe that the study now provides an important contribution to the literature in how the approach to enhanced surveillance in Romblon Province can provide guidance to developing similar methods for rabies control in other LMICs.

Thank you for your comments. We have gone through the text again to ensure readability, and paid particular attention to the areas highlighted (Lines 77-79, 126-127).

Reviewer #3 (Remarks to the Author):

3.1. Very well revised manuscript. However, there are still some points listed below that should be clarified or modified.

Thank you - we have tried to address all remaining concerns below.

Page 3, general comment about the introduction: Given the broad readership of Nature Communications, I suggest having the first paragraph being more general about the application of genomics and epidemiology for zoonotic disease research and then zooming into rabies in the 2nd paragraph of the introduction.

We have added an initial introductory paragraph on the application of genomic epidemiology to zoonotic disease research (Lines 49-54).

3.2. Page 4, line 83.

In terms of phylogeny informing transmission dynamics, etc. , it would be worth citing there some other instrumental works and reviews published by other groups. I listed below some examples:

- Mathematical modelling and phylodynamics for the study of dog rabies dynamics and control: A scoping review. By Layan et al., 2021. PMID: 34043640
- On the Use of Phylogeographic Inference to Infer the Dispersal History of Rabies Virus: A Review Study. By Nahata et al, 2021, PMID: 34452492
- Using phylogeographic approaches to analyse the dispersal history, velocity and direction of viral lineages - Application to rabies virus spread in Iran. By Dellicour et al., 2019 PMID: 31535448

Thank you for pointing to these important studies, which we had inadvertently overlooked. We now cite these on Line 93.

3.3. Page 6 line 137.

The authors refer to probable rabies cases in dogs “based on epidemiological evidence”. Could they provide details on how this probability was determined. Was it on clinical grounds? What is the rationale of this classification? Is this documented/validated in the literature? Need clarification on what “epidemiological evidence” refers too here.

We have now elaborated on the clinical signs reported for these animals that were used to classify them as probable rabies, see Lines 150-151.

3.4. Page 7, line 156-157.

If I am not mistaken, there are 31 samples tested by RDT and one of them is false negative. Therefore the sensitivity should be 96.8% instead of 95.7%.

There were 32 samples tested by RDT, and of these we get the following break down:

RDT-ve/DFAT-ve (true negative) = 10

RDT-ve/DFAT+ve (false negative) = 1

RDT+ve/DFAT+ve (true positive) = 21

RDT+ve/DFAT-ve (false positive) = 0

Sensitivity is calculated as: $TP/(TP+FN) = 21/22 = 95.5\%$

And specificity as: $TN/(TN+FP) = 10/10 = 100\%$

The small change is because when we went back to the data to double check our calculation, we realized we had not included a positive RDT which was conducted during a demonstration on animal sample collection (bringing the total number of RDTs to 32). This has now been amended in the text (Lines 162-163).

3.5. Page 7, line 164. Suggest changing to: "...the vaccination year AND THE TYPE OF VACCINE WERE unspecified." It is known that different vaccines may have different potencies. This should be relevant for rabies vaccines for veterinary use.

Thank you, we have clarified as suggested (Line 170).

3.6. Page 7, line 194. The authors mentioned "Information, Education and Communication (IEC" activities. These are not described in the M&M. Please describe or give a reference.

We have elaborated on the IEC undertaken in the methods as suggested (Line 201).

3.7. Page 8, line 211: I suggest adding: However, "lack of OFFICIAL" recognition..."

We appreciate your suggestion and have revised the text accordingly (Line 216).

3.8. Page 9: Fig2

Legend. Two human cases were not confirmed. Explain the basis on which these cases were considered as rabies cases. See similar comments on suspected animal cases above). Was this performed on clinical grounds, verbal autopsy...?

We have revised the legend to include the following explanation: "...this includes 2 human deaths from Romblon Island in 2020 that were diagnosed based on clinical signs but not confirmed so were not included in official government statistics to preserve the province's rabies-free status" (Lines 229-231).

3.9. Fig2a. There are many human deaths reported. It would be nice to separate those that were laboratory confirmed from those that are only suspected cases.

Human rabies deaths in the Philippines have historically been confirmed through clinical signs and outcome, which is why probable cases are considered even if not laboratory confirmed. Because of this, we have included further details in the legend emphasizing that only 2023 human rabies deaths were laboratory confirmed as a result of the recent implementation of genomic surveillance (Lines 229-231).

3.10. Page 11. Lines 269 to 293. There is large uncertainty on the introduction time (Jul2020 to Jun 2022 for cluster 1 and from Sep 1985-Jul 1994 for cluster 2. Why is the uncertainty so important? At least the authors should mention and discuss this

weakness. Further how was then the detection delay calculated considering these uncertainties? How do these uncertainties impact the simulation estimating the undetected cases? This calculation is not described in the M&M.

The uncertainty in the inferred introduction dates are due to the relatively few genomes available from which to calculate the timing of each cluster's MRCA. For cluster 2, we only have a single human case that is only distantly related to any previously sampled genomes. We explain that we were unable to resolve timings because of this undersampled diversity in the phylogeny (Lines 285-287, . For cluster 1, the situation is more complicated. The 3 transmission chains observed could either have each emerged through independent introductions all coming from the same foci, or could have been due to local circulation and divergence prior to detection. Without more genomes from this early stage of the outbreak (either from Tablas Island or from the inferred source, Bulacan province) we are unable to distinguish these scenarios. We have tried to make this clearer in the discussion as a limitation of the study (Lines 445-448).

The uncertainty in the introduction dates was not taken into account in the simulations of undetected cases, though we constrained the simulations to only those with plausible epidemiological dynamics (details reported in the methods, see Line 608-613). We discuss methodological limitations and are actively investigating how to improve methods for such inference, so as to provide more rigorous estimates for this and other such outbreaks in future (Lines 454-456).

3.11. Page 13, Fig. 4: Panel B and C do not match with the legend. Need to switch them around. Additionally, many thanks for adding the Supplementary Fig. S4, that is a great addition and makes the approach a lot clearer. However, a minor suggestion would be to increase the size of the panels and make the text less blurry in the figure. I found it hard to read.

Thanks to your observation, we have fixed the legend (Lines 329-332), with the descriptions now corresponding to their respective panels. We apologize for the blurry appearance of the figure as it has been automatically resized in-document to fit within the margins. We have ensured that the raw file has retained its high resolution and its text is clear and legible.

3.12. Page 13, line 335. Given that there is no causal effect between rabies introductions and decreased vaccination coverage associated with the COVID-19 pandemic, I suggest the authors tone down this argument, particularly in the abstract, because it implies it is a main finding of the study when it is more of speculation. Additionally, here the authors indicate that long-distance human-mediated translocations could be a contributing factor, but do not mention it in the abstract. Along those lines, have the authors considered that an increase in rabies cases post-pandemic could be linked simply to greater surveillance with the introduction of the

IBCM program during that time? Perhaps worth discussing in the Discussion. I also think there needs to be a discussion about the difference in vaccination coverage pre- vs post-pandemic (pre- ranging from 18-38.6% vs post- ranging from 0-24.2%) which arguably may not be different enough to influence rabies transmission in this system.

Thank you for your suggestions. We have revised the abstract to highlight human-mediated translocations as the source of introductions and toned down the point about suspended dog vaccination during the pandemic facilitating spread (Lines 35-37)

In the discussion we also added statements drawing attention to the poor vaccination coverage pre- and post-pandemic failing to interrupt rabies transmission (Lines 465-468).

With regards to our findings related to pandemic-induced suspension of dog vaccination, we now discuss the following:

- *The increase in human rabies cases nationwide post-pandemic (Lines 352-355)*
- *The lack of positive animal rabies cases among submissions in Romblon from 2017-2020 (Lines 379-382)*
- *The introduction of rabies into multiple rabies-free provinces post-pandemic (not just Romblon) even without IBCM implementation in those areas (Lines 359-360, 472-474).*

3.13. Page 14, line 343 and Page 16 line 454. It might be difficult for the reader to understand why 60% of vaccination rate is mentioned on page 14 and 70% on page 16. Please explain or homogenize.

The statement has been adjusted for clarity and consistency, and is now written as "60-70%" in accordance with the cited literature (Lines 348-350).

3.14. Page 14, line 375. To describe the use of RTD, the authors refer to 3 publications. It would worth to cite the original publication from 2016 who describe the use and the interest of this test in developing countries: Validation of a Rapid Rabies Diagnostic Tool for Field Surveillance in Developing Countries by Léchenne et al., 2016 PMID: 27706156.

Thank you, we now cite this reference as suggested (Line 386).

3.15. Page 15, line 393. Mentioning other emerging diseases seems a little out of context here since most of the manuscript is focused on rabies only. Suggest saying more about other diseases earlier or cutting this.

We have removed the sentence as suggested, since Lines 493-496 already establish the potential use of genomic surveillance for outbreaks of other diseases.

3.16. Page 17, line 475. Border control measures are not addressed in the is paper. Please give a reference dealing with this matter.

We have now cited a reference describing current Philippine border control measures for rabies (Line 490).

3.17. Page 18, Line 519. It would be nice to cite here (see my previous comment): Validation of a Rapid Rabies Diagnostic Tool for Field Surveillance in Developing Countries by Léchenne et al., 2016, PMID: 27706156.

Now cited (Line 535).

3.18. Page 18, line 526. Saliva and nuchal skin biopsies are not commonly used for the laboratory diagnosis of infectious diseases. Please explain as this is not described in the M&M. Give a reference. I suggest some examples below:

- A reliable diagnosis of human rabies based on analysis of skin biopsy specimens by Dacheux et al., 2008, PMID: 18937576
- Diagnostic tests for human rabies by Dacheux et al., 2018 PMID: 30747123

Thanks for these suggestions. We have now referenced them and have elaborated on the clinical and laboratory diagnosis of human cases as detailed earlier in our response (Lines 541-544).

3.19. Page 19, lines 539-542: Suggest adding a reference to back-up this approach.

We have added a reference as suggested (Line 555).

Reviewer #4 (Remarks to the Author):

Thank you for supporting this review.